# MotionBooth: Motion-Aware Customized Text-to-Video Generation

Jianzong Wu[1,3], Xiangtai Li[2,3] [†], Yanhong Zeng[3], Jiangning Zhang[4], Qianyu Zhou[5],
Yining Li[3], Kai Chen[3], Yunhai Tong[1]

[1]PKU [2]S-Lab, NTU [3]Shanghai AI Laboratory [4]ZJU [5]SJTU

**Project Page:** https://jianzongwu.github.io/projects/motionbooth
**Code:** https://github.com/jianzongwu/MotionBooth
*Email: jzwu@stu.pku.edu.cn, xiangtai94@gmail.com*

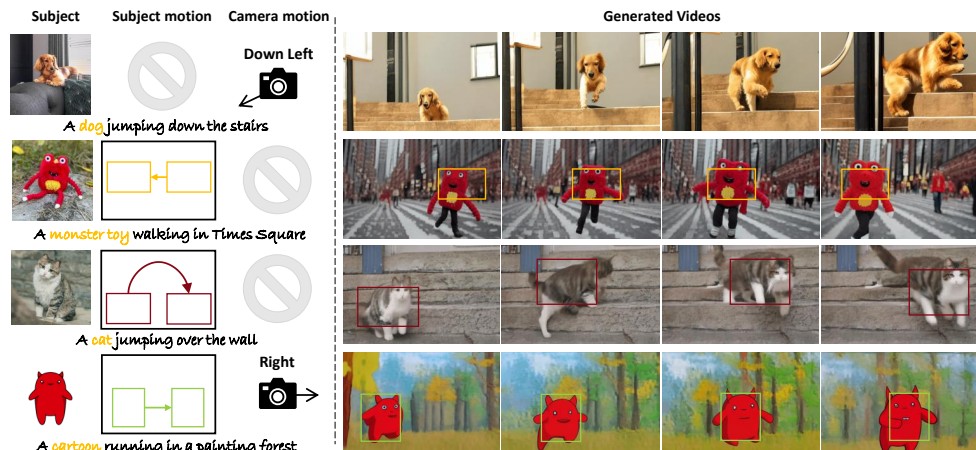

Figure 1: Motion-aware customized video generation results of MotionBooth. Our method animates a customized object with controllable subject and camera motions.

## Abstract

In this work, we present MotionBooth, an innovative framework designed for animating customized subjects with precise control over both object and camera movements. By leveraging a few images of a specific object, we efficiently fine-tune a text-to-video model to capture the object's shape and attributes accurately. Our approach presents subject region loss and video preservation loss to enhance the subject's learning performance, along with a subject token cross-attention loss to integrate the customized subject with motion control signals. Additionally, we propose training-free techniques for managing subject and camera motions during inference. In particular, we utilize cross-attention map manipulation to govern subject motion and introduce a novel latent shift module for camera movement control as well. MotionBooth excels in preserving the appearance of subjects while simultaneously controlling the motions in generated videos. Extensive quantitative and qualitative evaluations demonstrate the superiority and effectiveness of our method. Models and codes will be made publicly available.

---

Work done when Jianzong is an intern at Shanghai AI Laboratory. [†]: Project Lead. Corresponding: Xiangtai Li and Yunhai Tong.

38th Conference on Neural Information Processing Systems (NeurIPS 2024).

# 1 Introduction

Generating videos for customized subjects, such as specific scenarios involving a particular dog's type or appearance, has gained research attention [55, 25, 42]. This customized generation field originated from text-to-image (T2I) generation methods, which learn a subject's appearance from a few images and generate diverse images of that subject [12, 42, 30]. Following them, subject-driven text-to-video (T2V) generation has seen increasing interest, which has found a wide range of applications in personal shorts or film production [55, 25, 42, 57, 13]. Can you imagine your toy riding along the road from a distance to the camera or your pet dog dancing on the street from the left to the right? However, rendering such lovely imaginary videos is a challenging task. It often involves subject learning and motion injection while maintaining the generative capability to generate diverse scenes. Notably, VideoBooth [25] trains an image encoder to embed the subject's appearance into the model, generating a short clip of the subject. However, the generated videos often display minimal or missing motion, resembling a "moving image." This approach underutilizes the motion diversity of pre-trained T2V models. Another line of works [57, 61, 13] fine-tunes the customized model on specific videos, requiring motion learning for each specific camera or subject motion type. Their pipelines restrict the type of motion and require fine-tuning a new adapter for each motion type, which is inconvenient and computationally expensive.

The key lies in the conflict between *subject learning* and *video motion preservation*. During subject learning, training on limited images of the specific subject significantly shifts the distribution of the base T2V model, leading to significant degradation (e.g., blurred backgrounds and static video). Therefore, existing methods often need additional motion learning for specific motion control. In this paper, we argue that the base T2V model already has diverse motion prior, and the key is to preserve video capability during subject learning and digging out the motion control during inference.

To ensure subject-driven video generation with universal and precise motion control, we present MotionBooth, which can perform **motion-aware customized video generation**. The videos generated by MotionBooth are illustrated in Fig. 1. MotionBooth can take any combination of subject, subject motion, and camera motion as inputs and generate diverse videos, maintaining quality on par with pre-trained T2V models.

MotionBooth learns subjects without hurting video generation capability, enabling a training-free motion injection for subject-driven video generation. First, during subject learning, we introduce subject region loss and video preservation loss, which enhance both subject fidelity and video quality. In addition, we present a subject token cross-attention loss to connect the customized subject with motion control signals. During inference, we propose training-free techniques to control the camera and subject motion. We directly manipulate the cross-attention maps to control the subject motion. We also propose a novel latent shift module to govern the camera movement. It shifts the noised latent to move the camera pose. Through quantitative and qualitative experiments, we demonstrate the superiority and effectiveness of the proposed motion control methods, and they can be applied to different base T2V models without further tuning.

Our contributions are summarized as follows: **1)** We propose a unified framework, MotionBooth, for motion-aware customized video generation. To our knowledge, this is the first framework capable of generating diverse videos by combining customized subjects, subject motions, and camera movements as input. **2)** We propose a novel loss-augmented training architecture for subject learning. This includes subject region loss, video preservation loss, and subject token cross-attention loss, significantly enhancing subject fidelity and video quality. **3)** We develop innovative, training-free methods for controlling subject and camera motions. Extensive experiments demonstrate that MotionBooth outperforms existing state-of-the-art video generation models.

# 2 Related Work

**Text-to-video generation.** T2V generation leverages deep learning models to interpret text input and generate corresponding video content. It builds upon earlier breakthroughs in text-to-image generation [44, 41, 19, 21, 37, 48, 60, 62] but introduces more complex dynamics by incorporating motion and time [46, 20, 18, 2, 68, 59]. Recent advancements particularly leverage diffusion-based architectures. Notable models such as ModelScopeT2V [51] and LaVie [54] integrate temporal layers within spatial frameworks. VideoCrafter1 [6] and VideoCrafter2 [7] address the scarcity of video

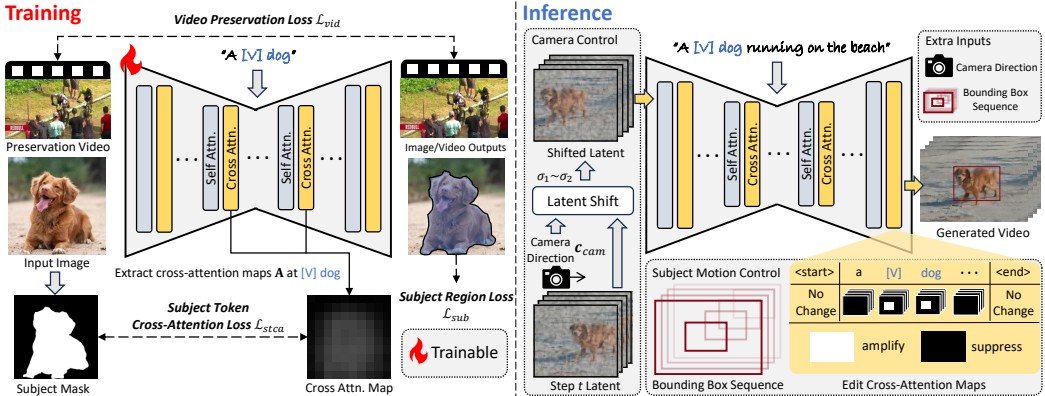

Figure 2: The overall pipeline of MotionBooth. We first fine-tune a T2V model on the subject. This procedure incorporates subject region loss, video preservation loss, and subject token cross-attention loss. During inference, we control the camera movement with a novel latent shift module. At the same time, we manipulate the cross-attention maps to govern the subject motion.

data by utilizing high-quality image datasets. Latte [36] and W.A.L.T [14] adopt Transformers as backbones [49]. VideoPoet [29] explores generating videos autoregressively to produce consistent long videos. Recent Sora [3] excels in generating videos with impressive quality, stable consistency, and varied motion. Despite these advancements, controlling video content through text alone remains challenging, highlighting a continuing need for research into more refined control signals.

**Customized generation.** Generating images and videos with customized subjects is attracting growing interest. Most works concentrate on learning a specific subject with a few images from the same subject [23, 8, 10, 43, 47, 39] or specific domains [15, 16, 50]. Textual Inversion [12] proposes to train a new word to capture the feature of an object. In contrast, DreamBooth [42] fine-tunes the whole U-Net, resulting in a better IP preservation ability. Following them, many works explore more challenging tasks, such as customizing multiple objects [30, 55, 33, 5], developing common subject adapter [58, 25, 65, 11, 67], and simultaneously controlling their positions [11, 33]. However, the customization of video models from a few images often results in overfitting. The models fail to incorporate significant motion dynamics. A recent work, DreamVideo [57], addresses this by learning specific motion types from video data. Yet, this method is restricted to pre-defined motion types and lacks the flexibility of text-driven input. In contrast, our work introduces MotionBooth to control both the subject and camera motions without needing pre-defined motion prototypes.

**Motion-aware video generation.** Recent works explore incorporating explicit motion control in video generation. This includes camera and object motions. To control camera motion, existing works like AnimateDiff [13], VideoComposer [53], CameraCtrl [17], Direct-A-Video [66], and MotionCtrl [56] design specific modules to encode the camera movement or trajectory. These models usually rely on training on large-scale datasets [1, 9], leading to high computational costs. In contrast, our MotionBooth framework builds a training-free camera motion module that can be easily integrated with any T2V model, eliminating the need for re-training. For object motion control, recent works [63, 31, 32, 24, 66, 4, 69, 27, 22] propose effective methods to manipulate attention values during the inference stage. Inspired by these approaches, we connect subject text tokens to the subject position using a subject token cross-attention loss. This allows for straightforward control over the motion of a customized object by adjusting cross-attention values.

# 3 Method

## 3.1 Overview

**Task formulation.** We focus on generating motion-aware videos featured by a customized subject. To customize video subjects, we fine-tune the T2V model on a specific subject. This process can be accomplished with just a few (typically 3-5) images of the same subject. During inference, the fine-tuned model generates motion-aware videos of the subject. The motion encompasses both camera and subject movements, which are freely defined by the user. For camera motion, the user

inputs the horizontal and vertical camera movement ratios, denoted as $\mathbf{c}_{cam} = [c_x, c_y]$. For subject motion, the user provides a bounding box sequence $[\mathbf{B}_1, \mathbf{B}_2, ..., \mathbf{B}_L]$ to indicate the desired positions of the subject, where $L$ represents the video length. Each bounding box specifies the x-y coordinates of the top-left and bottom-right points for each frame. By incorporating these conditional inputs, the model is expected to generate videos that include a specific subject, along with predefined camera movements and subject motions.

**Overall pipeline.** The overall pipeline of MotionBooth is illustrated in Fig. 2. During the training stage, MotionBooth learns the appearance of the given subject by fine-tuning the T2V model. To prevent overfitting, we introduce video preservation loss and subject region loss in Section 3.2. Additionally, we propose a subject token cross-attention (STCA) loss in Section 3.2 to explicitly connect the subject tokens with the subject's position on cross-attention maps, facilitating the control of subject motion. Camera and subject motion control are performed during the inference stage. We manipulate the cross-attention maps by amplifying the subject tokens and their corresponding regions while suppressing other tokens in Section 3.3. This ensures that the generated subjects appear in the desired positions. By training on the cross-attention map, the STCA loss enhances the subjects' motion control. For camera movement, we introduce a novel latent shift module to shift the noised latent directly, achieving smooth camera movement in the generated videos in Section 3.4.

## 3.2 Subject Learning

Given a few images of a subject, previous works have demonstrated that fine-tuning a diffusion model on these images can effectively learn the appearance of the subject [42, 23, 8, 10, 43, 47]. However, two significant challenges remain. First, due to the limited size of the dataset, the model quickly overfits the input images, including their backgrounds, within a few steps. This overfitting of the background impedes the generation of videos with diverse scenes, a problem also noted in previous works [42, 12]. Second, fine-tuning T2V models using images can impair the model's inherent ability to generate videos, leading to severe background degradation in the generated videos. To illustrate these issues, we

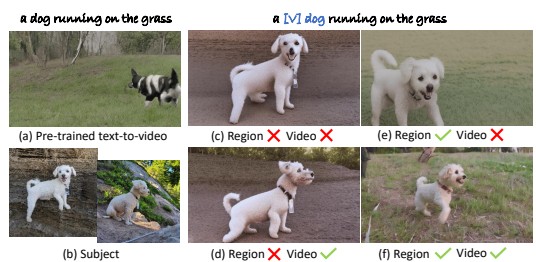

Figure 3: Case study on subject learning. "Region" indicates subject region loss. "Video" indicates video preservation loss. The images are extracted from generated videos.

conducted a toy experiment. As depicted in Fig. 3, without any modifications, the model overfits the background to the subject image. To address this, we propose computing the diffusion reconstruction loss solely within the subject region. However, even with this adjustment, the background in the generated videos remains over-smoothed. This degradation likely results from tuning a T2V model exclusively with images, which damages the model's original weights for video generation. To mitigate this, we propose incorporating video data as preservation data during the training process. Although training with video data but without subject region loss still suffers from overfitting, our approach, MotionBooth, can generate videos with detailed and diverse backgrounds.

**Preliminary.** T2V diffusion models learn to generate videos by reconstructing noise in a latent space [42, 30, 55, 12]. The input video is first encoded into a latent representation $\mathbf{z}_0$. Noise $\epsilon$ is added to this latent representation, resulting in a noised latent $\mathbf{z}_t$, where $t$ represents the timestamp. This process simulates the reverse process of a fixed-length Markov Chain [41]. The diffusion model $\epsilon_\theta$ is trained to predict this noise. The training loss, which is a reconstruction loss, is given by:

$$\mathcal{L} = \mathbb{E}_{\mathbf{z}, \epsilon \sim \mathcal{N}(\mathbf{0}, \mathbf{I}), t, \mathbf{c}} \left[ ||\epsilon - \epsilon_\theta(\mathbf{z}_t, \mathbf{c}, t)||_2^2 \right], \tag{1}$$

where $\mathbf{c}$ is the conditional input used in classifier-free guidance methods, which can be text or a reference image. During inference, a pure noise $\mathbf{z}_T$ is gradually denoised to a clean latent $\mathbf{z}_0'$, where $T$ is the length of the Markov Chain. The clean latent is then decoded back into RGB space to generate the video $\mathbf{X}'$.

**Subject region loss.** To address the challenge of overfitting backgrounds in training images, we propose a subject region loss. The core idea is to calculate the diffusion reconstruction loss exclusively

within the subject region, thereby preventing the model from learning the background. Specifically, we first extract the subject mask for each image. This can be done manually or through automatic methods, such as a segmentation model. In practice, we use SAM [28] to collect all the masks. The subject region loss is then calculated as follows:

$$\mathcal{L}_{sub} = \mathbb{E}_{\mathbf{z},\epsilon\sim\mathcal{N}(\mathbf{0},\mathbf{I}),t,\mathbf{c}} \left[||(\epsilon - \epsilon_\theta(\mathbf{z}_t, \mathbf{c}_i, t)) \cdot \mathbf{M}||_2^2\right], \qquad (2)$$

where $\mathbf{M}$ represents the binary masks for the training images. These masks are resized to the latent space to compute the dot product. $\mathbf{c}_i$ is a fixed sentence in the format "a [V] [class name]," where "[V]" is a rare token and "[class name]" is the class name of the subject [42]. We have found that with the subject region loss, the trained model effectively avoids the background overfitting problem.

**Video preservation loss.** Image customization datasets like DreamBooth [42] and CustomDiffusion [30] provide excellent examples of multiple images from the same subject. However, in the customized video generation task, directly fine-tuning the video diffusion model on images leads to significant background degradation. Intuitively, this image-based training process may harm the original knowledge embedded in video diffusion models. To address this, we introduce a video preservation loss designed to maintain video generation knowledge by joint training with video data. Unlike the class-specific preservation data used in previous works [42, 55], we utilize common videos with captions denoted as $\mathbf{c}_v$. Our experiments in Section 4 demonstrate that common videos are more effective for subject learning and preserving video generation capabilities. The loss function is formulated as follows:

$$\mathcal{L}_{vid} = \mathbb{E}_{\mathbf{z},\epsilon\sim\mathcal{N}(\mathbf{0},\mathbf{I}),t,\mathbf{c}} \left[||\epsilon - \epsilon_\theta(\mathbf{z}_t, \mathbf{c}_v, t)||_2^2\right]. \qquad (3)$$

**Subject token cross-attention loss.** To control the subject's motion, we directly manipulate the cross-attention maps during inference. Since we introduce a unique token, "[V]", in the training stage and associate it with the subject, we need to link this special token to the subject's position within the cross-attention maps. As illustrated in Fig. 4, fine-tuning the model does not effectively connect the unique token to the cross-attention maps. Therefore, we propose a Subject Token Cross-Attention (STCA) loss to guide this process explicitly. First, we extract the cross-attention map, $\mathbf{A}$, at the tokens "[V] [class name]". We then apply a Binary Cross-Entropy Loss to ensure that the corresponding attention map is larger at the subject's position and smaller outside this region. This process incorporates the subject mask and can be expressed as:

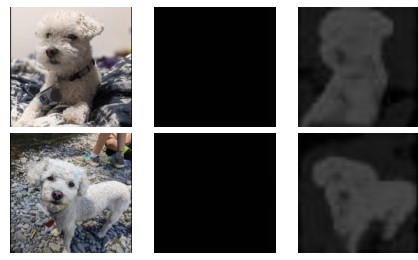

(a) Input Images (b) w/o STCA loss (c) w/ STCA loss

Figure 4: Case study on subject token cross-attention maps. (b) and (c) are visualization of cross-attention maps on tokens "[V]" and "dog".

$$\mathcal{L}_{stca} = -\left[\mathbf{M}\log(\mathbf{A}) + (1-\mathbf{M})\log(1-\mathbf{A})\right]. \qquad (4)$$

During training, the overall loss function is defined as:

$$\mathcal{L} = \mathcal{L}_{sub} + \lambda_1 \mathcal{L}_{vid} + \lambda_2 \mathcal{L}_{stca}, \qquad (5)$$

where $\lambda_1$ and $\lambda_2$ are hyperparameters that control the weights of the different loss components.

### 3.3 Subject Motion Control

We chose bounding boxes as the motion control signal for subjects because they are easy to draw and manipulate. In contrast, providing object masks for every frame is labor-intensive, requiring consideration of the subject's shape transformation between frames. In practice, we find that bounding boxes are sufficient for precisely controlling the positions of subjects. Previous works like GLIGEN [31] attempt to control object positions by training an extra condition module with large-scale image data. However, these training methods fix the models and cannot easily align with customized models fine-tuned for specific subjects. Therefore, we adopt an alternative approach that

directly edits the cross-attention maps during inference in a training-free manner [66, 27, 4]. This cross-attention editing method is plug-and-play and can be used with any customized model.

In cross-attention layers, the query features $\mathbf{Q}$ are extracted from the video latent and represent the vision features. The key and value features $\mathbf{K}$ and $\mathbf{V}$ are derived from input language tokens. The calculation process of the edited cross-attention layer can be formulated as follows:

$$\text{EditedCrossAttn}(\mathbf{Q}, \mathbf{K}, \mathbf{V}) = \text{Softmax}\left(\frac{\mathbf{Q}\mathbf{K}^\top}{\sqrt{d}} + \alpha\mathbf{S}\right)\mathbf{V}, \tag{6}$$

where $d$ is the feature dimension of $\mathbf{Q}$ and serves as a normalization term. $\frac{\mathbf{Q}\mathbf{K}^\top}{\sqrt{d}}$ is the normalized production between $\mathbf{Q}$ and $\mathbf{K}$, representing the attention scores between vision and language features. We manipulate the production by adding a new term $\alpha\mathbf{S}$, where $\mathbf{S}$ has positive values on the subject region provided in bounding boxes and large negative values outside the desired positions. $\alpha$ is a hyperparameter to control the editing strength. The editing matrix $\mathbf{S}$ is set as follows:

$$S_k[i,j] = \begin{cases} 1 - \frac{|\mathbf{B}_k|}{|\mathbf{Q}|}, & \text{if } i \in \mathbf{B}_k \text{ and } j \in \mathbf{P} \text{ and } t \geq \tau \\ 0, & \text{if } i \in \mathbf{B}_k \text{ and } j \in \mathbf{P} \text{ and } t < \tau \\ -\infty, & \text{otherwise} \end{cases} \tag{7}$$

where $i$, $j$, and $k$ indicate the vision token, language token, and frame indexes, respectively. $\mathbf{P}$ represents the indexes for subject language tokens in the text prompt. In this work, we choose "[V]" and "[class name]" as subject tokens. The SCTA loss in Section 3.2 binds the two tokens with cross-attention maps. $t$ is the denoising timestamp and $\tau$ is a hyperparameter defining a timestamp threshold. Since diffusion models tend to form the approximate object layout in earlier denoising steps and refine the details in later steps [63], we apply stronger attention amplification in earlier steps and no amplification in later steps. Note that attention suppression outside the bounding box regions persists throughout the generation. $|\mathbf{B}_k|$ and $|\mathbf{Q}|$ are the areas of the box and query, respectively. Following previous works [27, 66], smaller boxes should have larger amplifications, and we do not apply any editing on the <start> and <end> tokens.

**Discussion.** An important aspect is how the necessary information from other language tokens is integrated into the generated outputs, given that tokens such as verbs and background nouns are assigned minimal values. We propose that this information is extracted through the <start> and <end> tokens. Given that Transformer-based language encoders like CLIP [40] are typically trained on classification tasks, they often encode the overall context of a sentence into these special tokens. Thus, despite the suppression of other tokens during the softmax calculation, the model can still access relevant information about verbs, background elements, and other components necessary for the generation process. To support this explanation, we conducted an experiment in which we examined the softmax outputs using a naive text-to-video pipeline. The results showed that the <start> token consistently held the highest softmax value, close to 1, while the <end> token had the second-largest value. The remaining tokens, including those representing nouns, adjectives, verbs, and conjunctions, were distributed among the remaining softmax values.

### 3.4 Camera Movement Control

Simply editing the cross-attention map can efficiently control the motion of the subject. This suggests that the latent can be considered a "shrunk image," which maintains the same visual geographic distribution as the generated images. For camera movement control, an intuitive approach is to directly shift the noised latent during inference based on the camera movement signal $\mathbf{c}_{cam} = [c_x, c_y]$. The latent shift pipeline is illustrated in Table 2. The key challenge with this idea is filling in the missing parts caused by the latent shift (the question mark region in Step

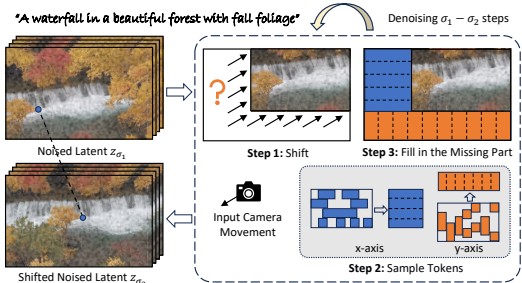

Figure 5: Illustration of camera movement control through shifting the noised latent.

1). To address this issue, we propose sampling tokens from the original noised latent and using them to fill the gap. This is based on the prior knowledge that when a camera moves in a video, the new scene it captures is semantically close to the previous one. For example, in a video with forest scenes, when the camera pans left, it is highly likely to capture more trees similar to those in the original scene. Another assumption is that in a normally angled video, a visual element is more likely to be semantically close to elements along the same x-axis or y-axis rather than other elements. For instance, in the waterfall video in Fig. 5, trees are at the top and bottom, spreading horizontally, while the waterfall spans the middle x-axis area. Experimentally, we over that sampling tokens horizontally and vertically provides better initialization and results in smoother video transitions. Randomly sampling tokens degrades the generated video quality. The latent shift process for timestamp $t$ can be formulated as follows:

$$
\begin{aligned}
\mathbf{h}_x &= \text{SampleHorizontal}(\mathbf{z}_t, \mathbf{B}, c_x), \\
\mathbf{h}_y &= \text{SampleVertical}(\mathbf{z}_t, \mathbf{B}, c_y), \\
\mathbf{z}_{\text{shift}} &= \text{Crop}(\text{Shift}(\mathbf{z}_t, c_x, c_y)), \\
\mathbf{z}_t &= \text{Fill}(\mathbf{z}_{\text{shift}}, \mathbf{h}_x, \mathbf{h}_y, c_x, c_y),
\end{aligned}
\tag{8}
$$

where $\mathbf{h}_x$ and $\mathbf{h}_y$ are sampled tokens along the x and y axes, respectively. $\text{Crop}(\cdot)$ removes the tokens outside the camera view after the shift. $\mathbf{B}$ is the subject bounding box. We filter out the tokens belonging to the subjects because they are not likely to occur in the new scenes. In addition, to avoid a drastic change in latent in one shift, we spread the latent shift over multiple timestamps, with each step only shifting a small number of tokens. Note that the latent shift needs to be applied after the subject's approximate layout is fixed but before the video details are completed. We set a pair of hyperparameters $\sigma_1$ and $\sigma_2$. The latent shift only applies in the timestamp range $[\sigma_1, \sigma_2]$.

## 4 Experiments

### 4.1 Experimental Setup

**Datasets.** For customization, we collect a total of 26 objects from DreamBooth [42] and CustomDiffusion [30]. These objects include pets, plushies, toys, cartoons, and vehicles. To evaluate camera and object motion control, we built a dataset containing 40 text-object motion pairs and 40 text-camera motion pairs, ensuring that the camera and object motion patterns are consistent with the text prompts. This dataset evaluates the videos generated for each subject in various scenarios and motions.

**Implementation details.** We train MotionBooth for 300 steps using the AdamW optimizer, with a learning rate of 5e-2 and a weight decay of 1e-2. We collect 500 preservation videos from the Panda-70M [9] training set, chosen randomly. Each batch consists of one batch for images and one for videos, with batch sizes equal to the number of training images and 1 for images and videos, respectively. The loss weight parameters $\lambda_1$ and $\lambda_2$ are set to 1.0 and 0.01. We use Zeroscope and LaVie as base models. During inference, we perform 50-step denoising using the DDIM scheduler and set the classifier-free guidance scale to 7.5. The generated videos are 576x320x24 and 512x320x16 for Zeroscope and LaVie, respectively. The training process finishes in around 10 minutes in a single NVIDIA A100 80G GPU. Additional implementation details can be found in Appendix A.1.

**Baselines.** Since we are pioneering motion-aware customized video generation, we compare our methods with closely related works, including DreamBooth [42], CustomVideo [55], and DreamVideo [57]. Dreambooth customizes subjects for text-to-image generation. We follow its practice with class preservation images and fine-tune T2V models for generating videos. CustomVideo is a recent video customizing method. We adopt its parameter-efficient training procedure. DreamVideo learns motion patterns from video data. To provide such data, we sample videos from Panda-70M, which are most relevant to the evaluation motions. Since these methods cannot control motions during inference, we apply our camera and object motion control technologies for a fair comparison. Additionally, we compare our camera control method with training-based methods, AnimateDiff [13] and CameraCtrl [17], focusing on camera motion control without subject customization. Since AnimateDiff is trained with only basic camera movement types and cannot take user-defined camera movement $\mathbf{c}_{cam} = [c_x, c_y]$ as input, we use the closest basic movement type for evaluation.

Table 1: **Quantitative comparison** for motion-aware customized video generation.

| T2V Model | Method | R-CLIP ↑ | R-DINO ↑ | CLIP-T ↑ | T-Cons. ↑ | Flow error ↓ |
|---|---|---|---|---|---|---|
| **Zeroscope** | DreamBooth [42] | 0.608 | 0.279 | 0.231 | 0.951 | 0.690 |
| | CustomVideo [55] | 0.657 | 0.267 | 0.245 | 0.955 | 0.516 |
| | DreamVideo [57] | 0.656 | 0.238 | **0.258** | 0.954 | 0.349 |
| | MotionBooth (Ours) | **0.667** | **0.306** | **0.258** | **0.958** | **0.252** |
| **LaVie** | DreamBooth [42] | 0.696 | 0.426 | 0.238 | 0.958 | 1.156 |
| | CustomVideo [55] | 0.634 | 0.189 | **0.248** | 0.911 | 1.055 |
| | DreamVideo [57] | 0.649 | 0.216 | 0.243 | 0.925 | 0.691 |
| | MotionBooth (Ours) | **0.712** | **0.472** | 0.247 | **0.962** | **0.332** |

Table 2: **Quantitative comparison** for camera movement control.

| Method | Module Weight Storage | FVD ↓ | CLIP-T ↑ | T-Cons. ↑ | Flow error ↓ |
|---|---|---|---|---|---|
| Text2Video-Zero (SD 1.5) [26] | [No Training] | 1821.72 | 0.248 | 0.904 | 1.854 |
| AnimateDiff [13] | 74M | 1515.82 | 0.245 | 0.925 | 1.683 |
| CameraCtrl [17] | 2.5G | 1468.53 | 0.237 | 0.939 | 0.807 |
| MotionCtrl [56] | 4.0G | 1109.45 | 0.236 | 0.935 | 0.872 |
| MotionBooth (Zeroscope) | [No Training] | 905.40 | **0.252** | 0.948 | **0.190** |
| MotionBooth (LaVie) | [No Training] | **723.26** | 0.241 | **0.963** | 0.296 |

**Evaluation metrics.** We evaluate motion-aware customized video generation from four aspects: region subject fidelity, temporal consistency, camera motion fidelity, and video quality. **1)** To ensure the subject is well-preserved and accurately generated in the specified motion, we introduce region CLIP similarity (R-CLIP) and region DINO similarity metrics (R-DINO). These metrics utilize the CLIP [40] and DINOv2 [38] models to compute the similarities between the subject images and frame regions indicated by bounding boxes. Additionally, we use CLIP image-text similarity (CLIP-T) to measure the similarity between entire frames and text prompts. **2)** We evaluate temporal consistency by computing CLIP image features between each consecutive frame. **3)** We use VideoFlow [45] to predict the optical flow of the generated videos. Then, we calculate the flow error by comparing the predicted flow with the ground-truth camera motion provided in the evaluation dataset. **4)** We randomly select 1000 videos from the MSRVTT dataset [64], predict their camera motion sequences with VideoFlow [45], and compute the FVD metric for camera motion control only.

### 4.2 Main Results

**Quantitative results.** We conduct quantitative comparisons with baseline models on both motion-aware customized video generation and camera movement control. The results for motion-aware customized video generation are shown in Table 1. The results demonstrate that MotionBooth outperforms all baselines on both Zeroscope and LaVie models, indicating that our proposed technologies can be extended to different T2V models. Thanks to the training-free architecture of subject and camera motion control methods, MotionBooth is expected to be adaptable to more open-sourced models in the future, such as Sora [3]. Notably, DreamVideo [57] achieves the second-best scores in T-Cons. and flow error, which aligns with our observation that incorporating video data as auxiliary training data enhances video generation performance. On the other hand, CustomVideo [55] shows inferior performance in R-DINO scores, indicating a poorer ability to generate subjects in given positions. This may be attributed to its approach of only fine-tuning the text embeddings and cross-attention layers of the diffusion models, which is insufficient for learning the subjects.

For camera movement control, we compare our method with two training-based methods, AnimateDiff [13] and CameraCtrl [17]. The results are shown in Table 2. Remarkably, MotionBooth achieves superior results compared to the two baselines with our training-free latent shift module. Specifically, MotionBooth outperforms the recent method CameraCtrl by 0.617, 0.015, and 0.009 in flow error, CLIP-T, and T-Cons. metrics with Zeroscope, and 0.511, 0.004, and 0.024 for the LaVie model. These results demonstrate that the latent shift method is simple yet effective.

**Qualitative results.** The qualitative comparison results for video generation with customized objects and controlled subject motions are presented in Fig. 6. Our observations reveal that MotionBooth excels in subject motion alignment, text prompt alignment, and overall video quality. In contrast,

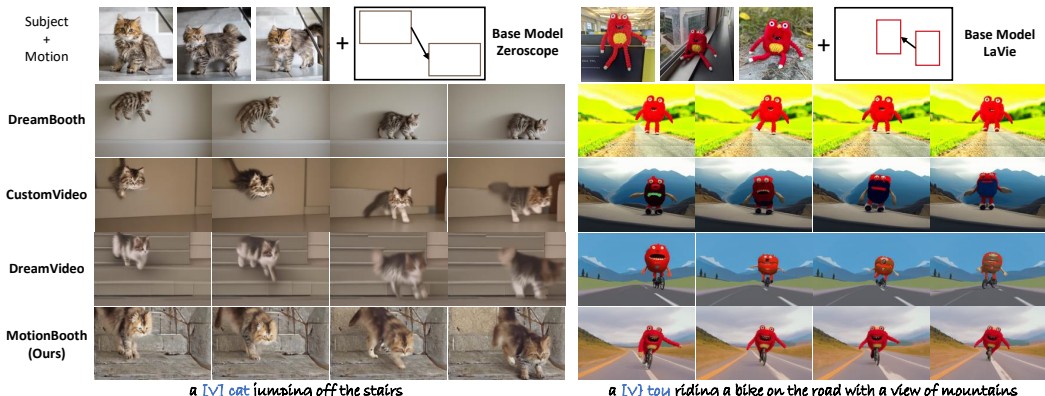

Figure 6: Qualitative comparison of customizing objects and controlling their motions.

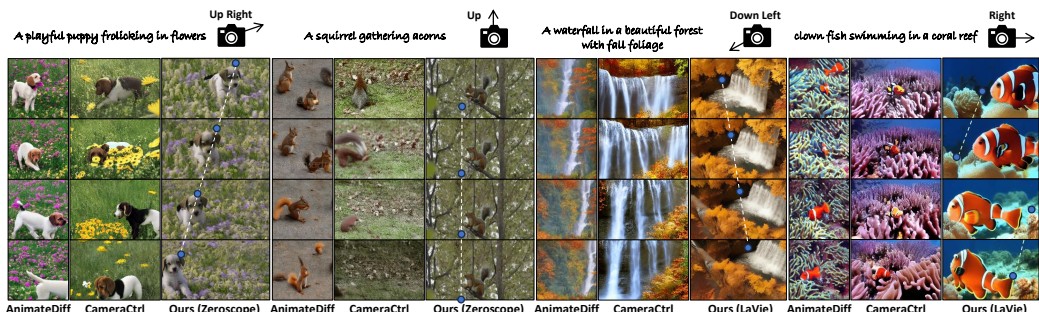

Figure 7: Qualitative comparison of camera motion control. Lines and points are used to help the readers track the camera movement more easily.

Table 3: **Ablation study for training technologies.** "mask" means subject region loss. "STCA" means subject token cross-attention loss. "video" means video preservation loss. "w/ class video" means utilizing class-specific videos in video preservation loss. The results are evaluated on LaVie.

| Method | R-CLIP ↑ | R-DINO ↑ | CLIP-T ↑ | T-Cons. ↑ | Flow error ↓ |
|---|---|---|---|---|---|
| w/o mask | 0.673 | 0.216 | 0.245 | 0.943 | 0.451 |
| w/o STCA | 0.710 | 0.453 | 0.244 | **0.962** | 0.363 |
| w/o video | 0.677 | 0.364 | 0.244 | 0.953 | 0.344 |
| w/ class video | 0.677 | 0.364 | 0.244 | 0.953 | 0.345 |
| MotionBooth | **0.712** | **0.472** | **0.247** | **0.962** | **0.332** |

DreamBooth and CustomVideo produce videos with vague backgrounds, highlighting that generated backgrounds deteriorate when training is conducted without video data. Additionally, CustomVideo and DreamVideo struggle to capture the subjects' appearances, likely because their approach tunes only part of the diffusion model, preventing the learning process from fully converging.

We also conduct qualitative experiments focused on camera movement control, with results shown in Fig. 7. AnimateDiff, limited to basic movements, does not support user-defined camera directions. Although the CameraCtrl method can accept user input, it generates videos with subpar aesthetics and objects that exhibit flash movements. In contrast, our MotionBooth model outperforms both the Zeroscope and Lavie models. The proposed latent method generates videos that adhere to user-defined camera movements while maintaining time consistency and high video quality.

### 4.3 Ablation Studies

**Training technologies.** We analyze the technologies proposed during the subject learning stage. The ablation results are shown in Table 3. Clearly, without the proposed modules, the quantitative

metrics drop accordingly. These results demonstrate that the proposed subject region loss, STCA loss, and video preservation loss are beneficial for subject learning and generating motion-aware customized videos. Specifically, the R-DINO metric decreases significantly by 0.256 without the subject region loss, highlighting its core contribution in filtering out image backgrounds during training. Additionally, the "w/ class video" experiment, which uses class-specific videos instead of randomly sampled common videos, yields worse results. This approach restricts the scenes and backgrounds in class-specific videos, hindering the models' ability to generalize effectively.

## 4.4 Human Preference Study

To evaluate our approach to understanding user preferences, we conducted a user study experiment. We collected 30 groups of videos generated by MotionBooth and baseline methods. We then asked 7 colleagues to select the best videos based on the following criteria: subject motion alignment, camera movement alignment, subject appearance alignment, and temporal consistency. For each group of videos, the annotators selected only the best one. For the subject appearance alignment, the annotators were provided with corresponding subject images. As

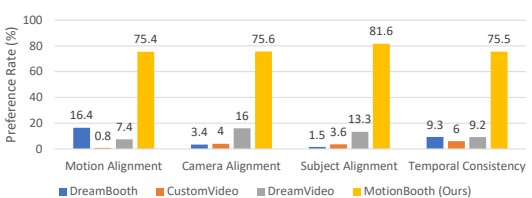

Figure 8: **Human preference study.** Our Motion-Booth achieves the best human preference scores in all the evaluation aspects.

shown in Fig. 8, MotionBooth was the most preferred method across all models and evaluation aspects, particularly in subject appearance alignment. These results further demonstrate the effectiveness of our method.

## 4.5 Limitations and Future Work

In Fig. 9, we illustrate several failure cases of MotionBooth. One significant limitation of Motion-Booth is its struggle with generating videos involving multiple objects. As shown in Fig. 9(a), the subject's appearance can sometimes merge with other objects, resulting in visually confusing outputs. This issue might be resolved by incorporating advanced training technologies for multiple subjects.

Another limitation is the model's capability to depict certain motions indicated by the text prompt. As depicted in Fig. 9(b), MotionBooth may fail to accurately represent motions that are unlikely to be performed by the subject. For example, it is hard to imagine a scene where a wolf plushie is riding a bike. These failure cases highlight the need for further improvement in the model's subject separation and motion understanding capabilities to enhance the realism and accuracy of the generated videos. Utilizing more powerful T2V models may eliminate these

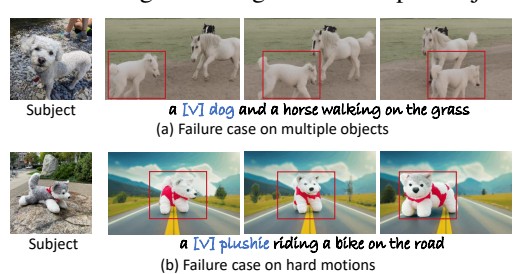

Figure 9: Failure cases of MotionBooth.

drawbacks. Future work could focus on refining these aspects to address the current limitations and provide more robust performance in complex scenarios.

## 5 Conclusion

This paper introduces MotionBooth, a novel framework for motion-aware, customized video generation. MotionBooth fine-tunes a T2V diffusion model to learn specific subjects, utilizing subject region loss to focus on the subject area. The training procedure incorporates video preservation data to prevent background degradation. Additionally, an STCA loss is designed to connect subject tokens with the cross-attention map. During inference, training-free technologies are proposed to control both subject and camera motion. Extensive experiments demonstrate the effectiveness and generalization ability of our method. In conclusion, MotionBooth can generate vivid videos with given subjects and controllable subject and camera motions.

**Acknowledgement.** This work is supported by the National Key Research and Development Program of China (No. 2023YFC3807600).

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

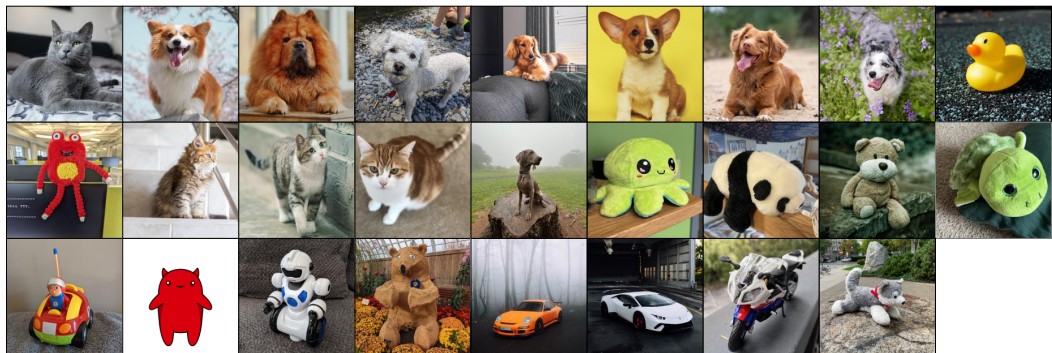

Figure 10: The evaluation dataset. We present one picture for each subject.

# A Appendix

**Overview.** The supplementary includes the following sections:

- **Appendix A.1.** Implementation details of the experiments.
- **Appendix A.2.** Discussions about our method and the differences with related methods.
- **Appendix A.3.** Comparison with more baselines.
- **Appendix A.4.** Ablation studies.
- **Appendix A.5.** Social impacts.
- **Appendix A.6.** More qualitative results.

**Video Demo.** We also present a video in a separate supplementary file, which shows the results in video format.

## A.1 Implementation Details

**Hyperparameters.** For the LaVie model, we set $\alpha = 10.0$, $\tau = 0.7\mathbf{T}$, $\sigma_1 = 0.8\mathbf{T}$, and $\sigma_2 = 0.6\mathbf{T}$. For the Zeroscope model, we set $\alpha = 10.0$, $\tau = 0.9\mathbf{T}$, $\sigma_1 = 0.9\mathbf{T}$, and $\sigma_2 = 0.7\mathbf{T}$.

**Evaluation dataset.** We collect a total of 26 subjects from DreamBooth [42] and CustomDiffusion [30]. We show one image for each subject in Fig. 10. The subjects contain a wide variety of types, including pets, plushie toys, cartoons, and vehicles, which can provide us with a thorough analysis of the model's effectiveness.

**User study interface.** We show the application interface for human preference study in Fig. 11. During user study, we ask the annotators to select the best video based on the question, e.g., "Which video do you think has the best temporal consistency?"

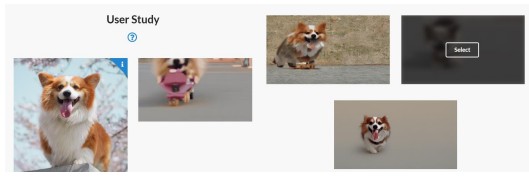

Figure 11: The application interface for user study.

**Pseudo-code of latent shift.** To present the latent shift module more clearly, we show the pseudo-code of the algorithm in Fig. 14. Our latent shift module can control the camera movement in videos in a training-free manner at minimal costs.

## A.2 Discussions

**Differences with related works.** Our proposed subject and camera motion control methods differ in several ways from previously established approaches, such as TrailBlazer [34], Directed Diffusion [35], Boximator [52], Motion-Zero [4], MotionCtrl [56], and Text2Video-Zero [26].

TrailBlazer [34] and Directed Diffusion [35] use a training-free approach for controlling object motion by manipulating the cross-attention maps. Specifically, TrailBlazer [34] adjusts both spatial and

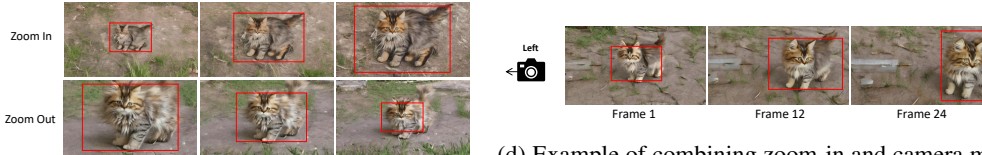

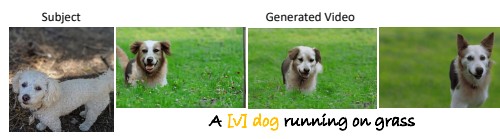

(a) Examples of enlarging or shrinking the subject bounding box.

(d) Example of combining zoom-in and camera motion control.

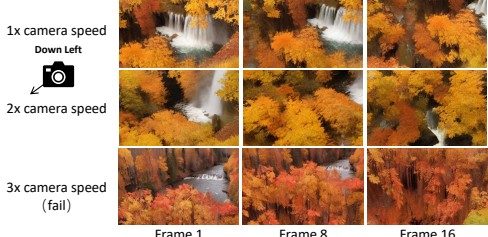

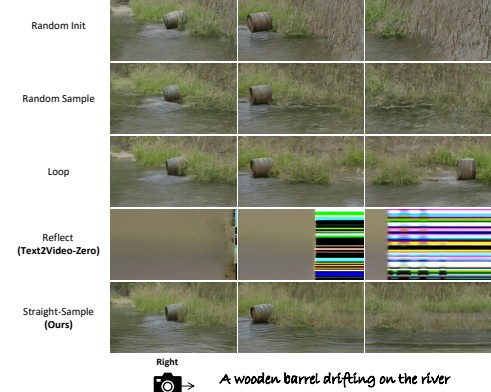

(b) Examples of gradually enlarging the camera motion speed.

(e) Example of masking images during training.

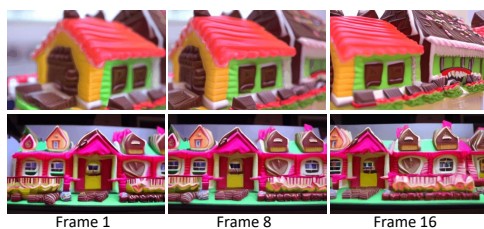

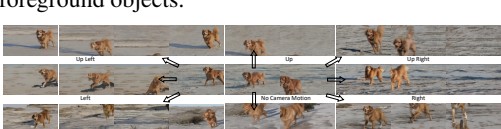

(c) Examples of camera motion control with large foreground objects.

(f) Ablations of the filling method in latent shift.

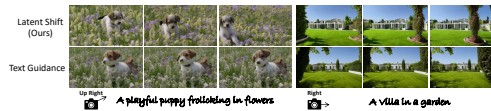

(g) Examples of controlling both subject and camera motion.

(h) Comparison of latent shift and text guidance to control camera motion.

Figure 12: **More qualitative results.**

temporal cross-attention by scaling the attention maps with a hyper-parameter of less than 1. Directed Diffusion [35] focuses on image generation in a similar manner. Our approach, in contrast, targets only the spatial cross-attention, setting attention values outside the object's bounding box to zero. This targeted manipulation simplifies implementation and significantly enhances the performance in generating motion-aware customized videos.

Boximator [52] relies on a training-based technique, requiring box coordinates as inputs to a newly trained self-attention layer. In comparison, our method is training-free, thus providing a more accessible and user-friendly solution for controlling subject and camera motion.

Motion-Zero [4] also operates without additional training, but it utilizes a test-time tuning technique to adjust the latent space using cross-attention map loss during the denoising process. This technique, however, increases the generation time and memory requirements considerably—from approximately 15 seconds to several minutes per video. Additionally, our experiments indicate that Motion-Zero often produces poor-quality outputs, with videos collapsing into unrecognizable elements. This outcome is likely due to the detrimental effects of test-time tuning on parameters and latent distributions in customized scenarios. In contrast, our method directly manipulates the cross-attention map, adding only 0.3 seconds to the generation process and consistently yielding more reliable visual outcomes.

MotionCtrl [56] uses a training-based approach that requires inputting point trajectories to control camera poses and object motion. Our method does not involve additional training, offering a simpler alternative for subject and camera motion control.

Text2Video-Zero [26] builds on a pre-trained text-to-image (T2I) model and extends it to video generation by utilizing consistent noise across frames. However, this approach is unsuitable for text-to-video (T2V) models, which typically use different noise for each frame. Additionally, Text2Video-Zero uses latent shifting for overall scene movement and mirrors latent information to fill in missing regions. In comparison, our approach employs random sampling along the x and y axes, resulting in more concise and coherent video generation for camera motion control.

We conducted quantitative experiments to evaluate the performance of these methods, as reported in Table 2 and Table 4b. Some methods are not included due to the unavailability of their code. Nonetheless, our results indicate that our method generally outperforms the other alternatives. In particular, we observed significant limitations in the test-time tuning strategy of Motion-Zero [4] when applied to customized video generation, which further emphasizes the strengths of our approach.

**Zoom-in/out effect with subject motion control.** Our subject motion control technique effectively manages changes in bounding box size, such as gradual enlargement or reduction. Specifically, enlarging the bounding box results in an appropriate scaling of the subject, creating a zoom-in effect in the generated video. Conversely, reducing the bounding box size produces a zoom-out effect. This zoom-in and zoom-out behavior is demonstrated in Fig. 12a, where examples illustrate these effects in response to bounding box adjustments. Fig. 12d shows an example of combining zoom-in and camera motion control, demonstrating the method's flexibility.

**Large camera motion speeds.** We evaluate the performance of our camera motion control technique under varying levels of camera movement intensity. As depicted in Fig. 12b, the method was first tested with a camera movement vector of $[c_x, c_y] = [-0.5, 0.45]$, corresponding to a movement of half the video width to the left and nearly half downward. Under these conditions, our method successfully managed the camera movement, producing accurate results. When the camera motion speed was doubled, the model continued to perform well, demonstrating its ability to handle high-speed scenarios. However, at a speed three times the initial setting, i.e., $[c_x, c_y] = [-1.5, 1.35]$, the method exhibited limitations, resulting in only a downward tiling effect. These findings indicate that while our method can effectively manage camera movements spanning the entire video width, its performance diminishes under extremely high-speed conditions.

**Camera motion control with large foreground objects.** We evaluate the camera control capability in scenes containing substantial foreground objects. The results, presented in Fig. 12c, show an experiment involving a scene dominated by a large candy house, which nearly occupies the entire frame. Despite the significant presence of the foreground object, our technique effectively managed to pan the camera to the right, suggesting that the method is capable of handling complex scenes with large foreground elements. This robustness is attributed to the method's reliance on latent shifts, which simultaneously move both foreground and background elements, thereby ensuring that the presence of large foreground objects does not significantly impair performance.

**Model efficiency.** Training our model by fine-tuning a specific subject takes approximately 10 minutes. For LaVie [54], the naive text-to-video (T2V) pipeline takes about 15.0 seconds per video. When incorporating subject control, the inference time is 15.3 seconds per video; with camera control, it increases to 20.6 seconds per video; and when applying both camera and subject control, the inference time is 21.5 seconds per video.

## A.3 Comparison with More Baselines

Table 4: **Comparison with More Baselines.**

(a) Comparison of the latent shift method and text guidance for camera motion control.

| Method | CLIP-T ↑ | T-Cons. ↑ | Flow error ↓ |
|---|---|---|---|
| Text Guidance | **0.256** | **0.957** | 0.416 |
| Latent Shift (Ours) | 0.252 | 0.948 | **0.190** |

(b) Comparison of subject motion control with more baselines.

| Method | R-CLIP | R-DINO | CLIP-T ↑ | T-Cons. ↑ |
|---|---|---|---|---|
| Motion-Zero [4] | - | - | Collapse | - |
| Directed Diffusion [35] | 0.668 | 0.242 | **0.260** | 0.954 |
| TrailBlazer [34] | 0.669 | 0.251 | 0.259 | 0.957 |
| Ours (Zeroscope) | **0.767** | **0.305** | 0.242 | **0.968** |
| Ours (LaVie) | 0.735 | 0.247 | 0.244 | 0.965 |

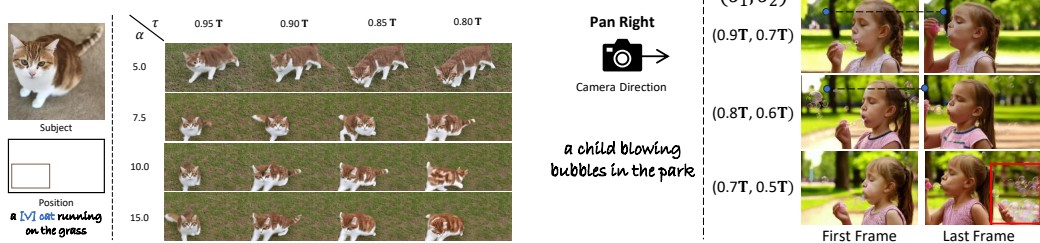

(a) Ablation study on subject motion control. Only the first frame is shown. Experiments on Zeroscope.

(b) Ablation study on latent shift. Experiments on LaVie. A higher $\sigma$ means an earlier denoising step.

Figure 13: **Ablation study on motion control hyperparameters.**

Table 5: **Ablation study for the number of video preservation data.**

| # Videos | R-CLIP ↑ | R-DINO ↑ | CLIP-T ↑ | T-Cons. ↑ | Flow error ↓ |
|---|---|---|---|---|---|
| 100 | 0.714 | 0.488 | 0.245 | 0.964 | 0.324 |
| 300 | 0.711 | 0.486 | 0.245 | 0.962 | 0.330 |
| 500 | 0.712 | 0.472 | 0.247 | 0.962 | 0.332 |
| 700 | 0.712 | 0.473 | 0.247 | 0.963 | 0.340 |
| 900 | 0.712 | 0.480 | 0.244 | 0.963 | 0.328 |

**Comparison with text guidance for camera motion control.** we compare our latent shift method with text guidance for controlling camera motion with the results presented in Table 4a and Fig. 12h. Specifically, we utilize simple text prompts, such as "camera pan right" and "camera pan up right," to influence camera movement. The generated videos reflect these instructions to some extent, especially for straightforward motions like panning to the right. However, we observe that such text prompts often lack sufficient specificity, particularly in terms of conveying critical details such as the speed or distance of the camera movement. As a result, this approach yielded sub-optimal outcomes when compared to our proposed method. This indicates that our approach offers a more precise and stable mechanism for controlling camera motion than is possible with text prompts alone.

**Comparison of subject motion control with more baselines.** Table 4b presents quantitative experiments comparing our method with several baseline approaches, including TrailBlazer [34], Directed Diffusion [35], and Motion-Zero [4]. The results demonstrate that our approach generally outperforms these alternatives.

## A.4    Ablation Studies

**Subject motion control hyperparameters.** We conduct ablation studies on the hyperparameter for subject motion control, with the results presented in Fig. 13a. We examined the effects of varying the factor $\alpha$ of $\mathbf{S}$ and the maximum cross-attention manipulation timestamp $\tau$. The findings indicate that increasing $\alpha$ and extending the controlling steps lead to stronger control. With lower control strengths, the subject does not appear in the desired position, or only part of its body aligns with the intended spot. Conversely, when the control strengths are too high, the generated subjects tend to appear unnaturally square in shape.

**Latent shift hyperparameters.** We experiment with the influence of $\sigma_1$ and $\sigma_2$ in the latent shift module. The results are shown in Fig. 13b. The results indicate that applying latent shift in the earlier steps of the denoising process results in incomplete camera movement, as evidenced by the trees in the background. Conversely, shifting the latent in the later steps degrades video quality and introduces artifacts, highlighted by the red boxes in the last row. Empirically, setting $\sigma_1$ and $\sigma_2$ to middle values provides optimal control over camera movement.

**Number of preservation videos.** We conduct an ablation study on the number of preservation videos. As shown in Table 5, ranging the preservation videos from 100 to 900 does not bring large changes to the quantitative scores. We conclude that the key is to use video data to preserve the video generation ability of the pre-trained T2V models. The number of video data can be flexible.

Table 6: **More ablation studies.**

(a) Ablation of controlling single motion type.

| T2V Model | Evaluation Set | R-CLIP↑ | R-DINO↑ | CLIP-T↑ | T-Cons.↑ | Flow error↓ |
|---|---|---|---|---|---|---|
| Zeroscope | Only Subject | **0.767** | 0.305 | 0.242 | **0.968** | - |
| | Only Camera | - | - | 0.252 | 0.948 | **0.190** |
| | Both | 0.667 | **0.306** | **0.258** | 0.958 | 0.252 |
| LaVie | Only Subject | **0.735** | 0.247 | 0.244 | **0.965** | - |
| | Only Camera | - | - | 0.241 | 0.963 | **0.296** |
| | Both | 0.712 | **0.472** | **0.247** | 0.962 | 0.332 |

(b) Ablation of masking the training images.

| Method | R-CLIP↑ | R-DINO↑ | CLIP-T↑ | T-Cons.↑ |
|---|---|---|---|---|
| Mask Image | 0.683 | 0.060 | 0.237 | 0.937 |
| Mask Loss (Ours) | **0.712** | **0.472** | **0.247** | **0.962** |

(c) Ablation of the latent filling method in latent shift.

| Method | CLIP-T↑ | T-Cons.↑ | Flow error↓ |
|---|---|---|---|
| Random Init | 0.235 | 0.922 | 0.212 |
| Random Sample | 0.248 | 0.954 | 0.185 |
| Loop | 0.252 | **0.956** | **0.168** |
| Reflect | - | Collapse | - |
| Ours | **0.252** | 0.948 | 0.190 |

**Controlling Single Motion Type.** We conduct an additional experiment focusing specifically on the scenario where only subject motion is controlled, and the results are summarized in Table 6a. Notably, we observe that the inclusion of both subject and camera motion controls leads to a slight decrease in some metrics. For instance, using the Zeroscope model, there is a 0.1 drop in the R-CLIP score and a 0.01 decrease in the T-Cons metric when both types of motion are controlled compared to the scenario where only subject motion is controlled. However, there are also instances where metrics such as R-DINO and CLIP-T show a slight improvement under combined control conditions. We consider this trade-off acceptable within the context of motion-aware customized video generation. Qualitative results in Fig. 12g show that our method can successfully handle controlling both subject and camera motion.

**Masking the training images.** We conduct an ablation study to evaluate the effect of masking the background regions directly in pixel space of the training images, compared to masking the diffusion loss. As illustrated in Fig. 12e, directly masking the training images significantly impairs the model's capacity to learn the subject effectively. The quantitative metrics shown in Table 6b indicate a marked decrease in performance when the training images are masked. We hypothesize that this decline arises from the unnatural distribution created by the masked images, which disrupts the learning process. Consequently, we conclude that masking the diffusion loss, rather than masking the training images directly, is a more effective strategy for preserving the integrity of the learned representations.

**Latent Filling Methods.** We test several latent filling approaches and present the qualitative results in Fig. 12f and the quantitative results in Table 6c.

*Random Init:* This method involves filling the hole with random values. Our experiments revealed that this technique leads to severe artifacts due to the disruption of the natural horizontal and vertical distribution of latent pixels, ultimately degrading the overall visual quality.

*Random Sample:* In this approach, values are randomly sampled in the latents. Similar to *Random Init*, this method produced significant artifacts, leading to poor visual quality in the generated video.

*Loop:* This method reuses the moved-out-of-scene values to fill the missing region, thereby creating a looping background effect. While this technique was found to preserve video quality better than the random methods, it introduced a limitation in terms of flexibility for camera movements, resulting in repetitive looping effects. Therefore, it is not suitable for more diverse camera controls.

*Reflect:* This approach is employed in Text2Video-Zero [26], where the missing region is filled by reflecting the surrounding content. However, in our Text-to-Video (T2V) scenario, this method collapsed, failing to maintain the desired visual quality.

The quantitative results presented in Table 6c corroborate these findings. *Random Init* and *Random Sample* lead to significant artifacts, whereas the *Loop* method provides better visual quality but at the cost of limiting camera movement diversity. The *Reflect* method, despite its success in other applications, did not yield satisfactory results in our T2V context.

In conclusion, our proposed *straight-sample* method consistently maintained high visual quality without introducing significant artifacts or limiting camera movement flexibility, demonstrating its superiority in maintaining the desired video fidelity.

## A.5 Social Impacts

**Positive societal impacts.** MotionBooth allows for precise control over customized subjects and camera movements in video generation, opening new avenues for artists, filmmakers, and content creators to produce unique and high-quality visual content without extensive resources or professional equipment.

**Potential negative societal impacts.** The ability to generate realistic customized videos could be misused to create deepfakes, leading to potential disinformation campaigns, privacy violations, and reputational damage. This risk is particularly significant in the context of political manipulation and social media. If the underlying models are trained on biased datasets, the generated content might reinforce harmful stereotypes or exclude certain groups. Ensuring diversity and fairness in training data is crucial to mitigate this risk.

**Mitigation strategies.** Developing and adhering to strict ethical guidelines for the use and dissemination of video generation technologies can help mitigate misuse. This includes implementing usage restrictions and promoting transparency about the generated content.

### A.6 More Qualitative Results

We show more qualitative results in Fig. 15.

```python
def shift_latent_one_step(latent, cx, cy, bbox, num_shift_steps):
    # latent: noised latent for timestamp t
    # cx: x-axis speed for camera movement
    # cy: y-axis speed for camera movement
    # bbox: the bounding box for the customized subject
    # num_shift_steps: σ₁ − σ₂, the total steps needed to complete latent shift

    # get the latent shape
    batch_size, channels, num_frames, height, width = latent.shape

    # divide the camera speed by num_shift_steps.
    # for each step, we move a part of the total distance
    sx = cx / num_shift_steps
    sy = cy / num_shift_steps

    for f in range(num_frames):
        # define latent shift distance for each frame
        sfx = int(sx * f)
        sfy = int(sy * f)

        # define a obj_mask to avoid sampling tokens within the subject region
        obj_mask = torch.ones_like(latent[0,0,f,:,:])
        obj_mask[bbox] = False

        # sampling tokens horizontally
        if sfx != 0:
            fill_x = torch.zeros_like(latent[:,:,f,:,:abs(sfx)])
            for i in range(height):
                included_indices = [x for x in range(0, width) if obj_mask[i,x]]
                sampled_indices = random_choice(included_indices, size=abs(sfx)
                fill_x[:,:,i,:] = latent[:,:,f,i,sampled_indices]
        # sampling tokens vertically
        if sfy != 0:
            fill_y = torch.zeros_like(latent[:,:,f,:abs(sfy),:])
            for j in range(width):
                included_indices = [y for y in range(0, height) if obj_mask[y,j]]
                sampled_indices = random_choice(included_indices, size=abs(sfy)
                fill_y[:,:,:,j] = latent[:,:,f,sampled_indices,j]

        # shift the original latent and fill in the hole with sampled tokens
        if sfx > 0:
            temp = latent[:,:,f,:,sfx:]
            latent[:,:,f,:,:] = torch.cat([temp, fill_x], dim=-1)
        elif sfx < 0:
            temp = latent[:,:,f,:,:sfx]
            latent[:,:,f,:,:] = torch.cat([fill_x, temp], dim=-1)
        if sfy > 0:
            temp = latent[:,:,f,sfy:,:]
            latent[:,:,f,:,:] = torch.cat([temp, fill_y], dim=-2)
        elif sfy < 0:
            temp = latent[:,:,f,:sfy,:]
            latent[:,:,f,:,:] = torch.cat([fill_y, temp], dim=-2)

    return latents
```

Figure 14: Pseudo-code of the latent shift algorithm.

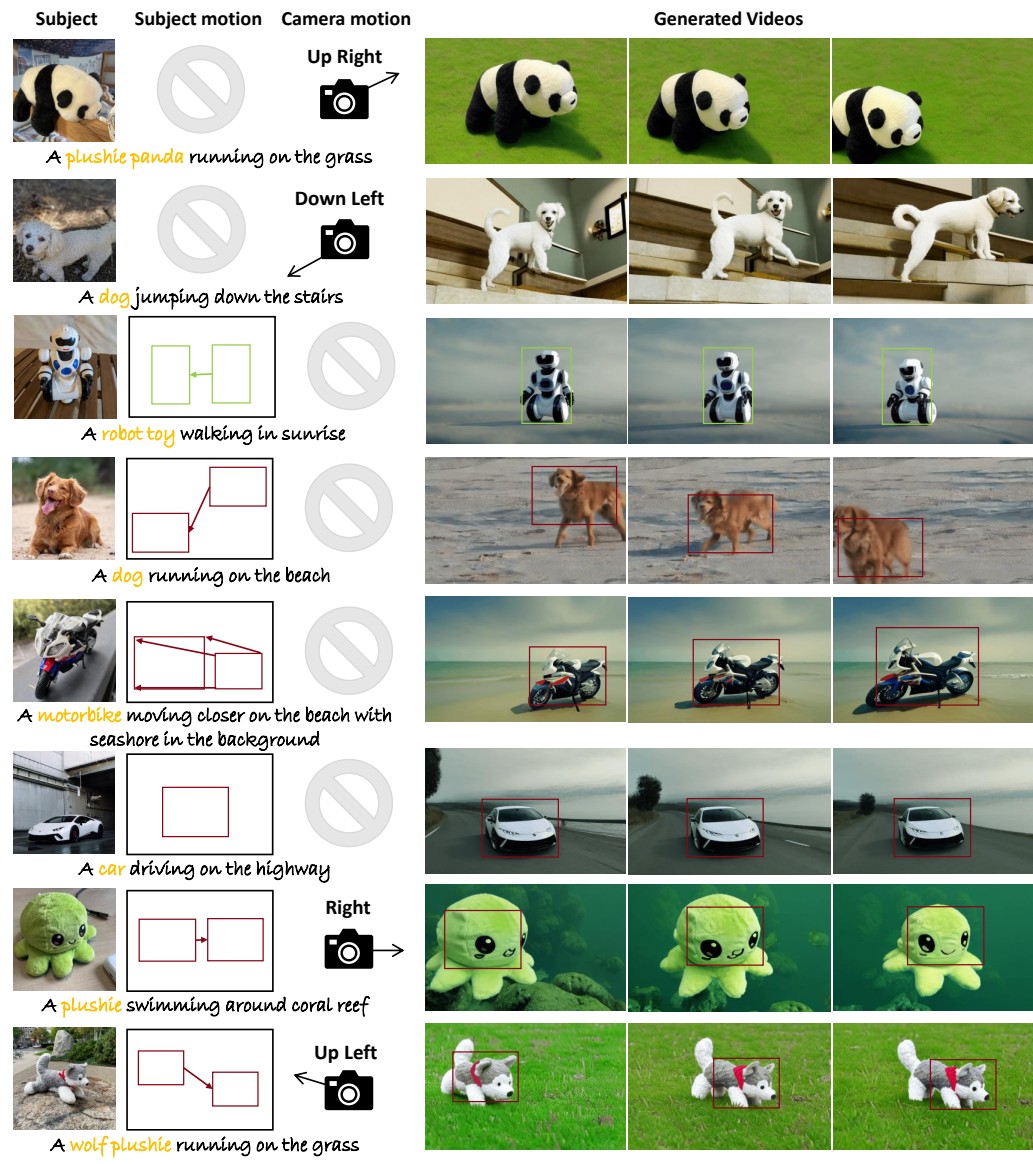

Figure 15: More qualitative results of our MotionBooth.

