# OpenReview forum: "MotionBooth: Motion-Aware Customized Text-to-Video Generation"
_NeurIPS.cc/2024/Conference — NeurIPS 2024 spotlight_

### Official Review · Reviewer_q2XJ · 2024-07-06

**Soundness:** 2
**Presentation:** 3
**Contribution:** 2
**Rating:** 3
**Confidence:** 5

**Summary:**

The authors propose MotionBooth, a method to fine-tune a pre-trained text-to-video model on a collection of images of a specific object to enable the model to generate controlled videos of that object. Fine-tuning incorporates three losses: diffusion loss on image data, restricted to the object's region; video preservation loss, i.e diffusion loss on sample video data, to prevent overfitting to static images; subject token cross-attention loss, to improve controllability at generation time. During inference the motion of the object is controlled via editing the cross-attention maps to amplify the attention to the object tokens within the object's bounding boxes. The camera is controlled via shifting the noised latent and filling the new regions with latents sampled from the background of the original noised latent.

**Strengths:**

The problem of controllable video generation is of high significance nowadays. The method proposed in the paper is relatively simple and the demonstrated visual results confirm the improved controllability of MotionBooth over prior work. The contributions in the paper are supported with well-designed figures aiding the clarity of the presentation. The limitations of the proposed method are discussed in the appendix.

**Weaknesses:**

1) **Novelty**:
    - Similar training-free object motion control was proposed in [1] and [2] for image and video models respectively, where cross-attention amplification within object's bounding box was used to control its location in the generated frame. Similar training-free camera motion control was used in [3], where a series of noised latent shifts were used to control the global scene and camera motion.
Discussion and comparisons with those works are missing in the paper.
    - Some missing references: [4, 5].

2) **Method**:
    - Based on the formula for subject motion control, the attention is suppressed (set to -infinity) for all query-key pairs other than the queries from the bounding box and the keys from the object tokens. This way it seems that the rest of the prompt is ignored. Moreover, it is stated in the paper that "the attention suppression outside the bounding box regions persists throughout the generation". Details clarifying this are missing in the paper.

3) **Evaluation**:
    - The paper lacks quantitative evaluation. Comparing CLIP features between the frames doesn't measure temporal consistency. Metrics like FVD, or optical flow warpping error could be better choices for this purpose.
    - Details on how exactly the metrics are calculated are missing. Formal definitions of newly introduced metrics are required for better understanding. E.g. how exactly is the flow error calculated in the case when not all the motion except for the object's motion corresponds to camera motion? Having those would help with understanding the significance of the improvements reported in the tables.
    - The evaluation dataset consists of limited number of different object classes, mostly dogs, cats and toys. Evaluation on more general collection of object classes would better support the claims made in the paper.
    - More ablation studies would better illustrate the contributions. E.g. what would happen if the missing region in the shifted noised latent was not filled with samples from the same noised latent? Or if camera control like in [3] was used instead?

4) **Results**:
    - Figure 14 shows that bounding boxes often don't fully contain the object of interest. Given that the attention is suppressed outside of bounding boxes, this looks like a flaw in the proposed control and needs to be investigated.

[1] Ma et al., Directed diffusion: Direct control of object placement through attention guidance, AAAI 2024.

[2] Ma et al., TrailBlazer: Trajectory Control for Diffusion-Based Video Generation, arxiv 2024.

[3] Khachatryan et al., Text2Video-Zero: Text-to-Image Diffusion Models are Zero-Shot Video Generators, ICCV 2023.

[4] Wang et al., Boximator: Generating rich and controllable motions for video synthesis, arxiv 2024.

[5] Chen et al. Motion-Zero: Zero-Shot Moving Object Control Framework for Diffusion-Based
Video Generation, arxiv 2024.

**Questions:**

1) Could the authors discuss the connection between the proposed MotionBooth and the prior work with similar training-free controls? Some ablation studies would be ideal.

2) Could the authors provide more details about their subject motion control attention suppression technique? Especially regarding the issue with attention to the prompt tokens other than the subject language tokens.

3) The authors claim that their fine-tuning is efficient (line 3 in the abstract). Could the authors provide more details on this? E.g. what is the training time to adapt the model to a new subject?

**Limitations:**

The authors have adequately addressed the limitation and the potential negative societal impact of their work.

---

> ### Author Rebuttal · Authors · 2024-08-07
>
> Thanks very much for the review! Here are our replies.
>
> **Note: Please refer to the one-page PDF for the mentioned figures and tables with an "R." in their names.**
>
>
> # Novelty of our method
>
> Thanks for the suggestion. We appreciate that most reviewers have acknowledged the novelty of our method and we will add the missing citations in the final version.
>
> Due to the response word limit, please see the general response for comparison to related methods.
>
>
> # Ablations of the latent shift filling method
>
> To address the reviewer's question regarding additional ablation studies, we have conducted experiments to explore different methods for filling the missing region in the shifted noised latent. We present the qualitative results in `Figure R.1 (f)` and the quantitative results in `Table R.1 (c)`.
>
> In our experiments, we tested several approaches:
>
> 1. Random Init: Filling the hole with random values. This method resulted in severe artifacts due to the disruption of the natural horizontal and vertical distribution of latent pixels.
> 2. Random Sample: Randomly sampling values in the latents. Similar to Random Init, this approach also produced significant artifacts and degraded the visual quality.
> 3. Loop: This method involves using the moved-out-of-scene values to fill the hole, creating a looping background effect. While this technique preserves video quality better than random initialization or sampling, it limits the flexibility of camera movements by producing only looping effects. Thus, it is not suitable for more varied camera controls.
> 4. Reflect: This is the method used in Text2Video-Zero, where the missing region is filled by reflecting the surrounding content. However, in our Text-to-Video (T2V) situation, this approach resulted in the method collapsing, failing to maintain the desired quality.
>
> The quantitative results in `Table R.1 (c)` corroborate these observations, showing that Random Init and Random Sample lead to significant artifacts, while Loop provides better quality but lacks flexibility. The Reflect method, although effective in other contexts, did not perform well in our scenario.
>
> In conclusion, our straight-sample method consistently maintained high visual quality without introducing significant artifacts or limiting camera movement flexibility.
>
>
> # Details about attention suppresion
>
> Thank you for your insightful question. In our approach, we manipulate the dot product between the query matrix $\mathbf{Q}$ and the key matrix $\mathbf{K}$. The final attention map is then calculated using a Softmax function across all tokens.
>
> The amplification of the subject tokens is during the earlier denoising steps.
>
> In the later steps, we suppress all the tokens, including the subject tokens all to $-\infty$. This does not completely ignore any tokens due to the nature of the Softmax function, which normalizes the scores across all tokens.
>
> Our qualitative results, as presented in the paper and the accompanying one-page PDF, demonstrate that the generated videos do incorporate other aspects of the prompt, such as verbs or background elements. This shows that while the subject tokens receive amplified attention, the rest of the prompt tokens are still considered, influencing the overall output.
>
>
> # Video metrics
>
> Thanks for the feedback. Due to the word limit, please refer to the general response.
>
>
> # Flow error metrics calculation
>
> We exclude the subject region when calculating flow error, so the flow error only accounts to camera motion control.
>
> The flow error $E_{flow}$ for each frame can be expressed as:
>
> $$
> E_{flow} = \frac{1}{|R|} \sum_{p \in R} \| \mathbf{F}_p - {\mathbf{C}} \|,
> $$
>
> where $\mathbf{F}_p$ is the predicted flow vector at pixel $p$. ${\mathbf{C}}$ is the ground truth camera motion condition. $R$ represents the region outside the subject bounding box. $|R|$ is the number of pixels in region $R$.
>
>
> # Evaluation datasets
>
> Thank you for your feedback. We acknowledge that current open-sourced, state-of-the-art T2V models, such as LaVie and Zeroscope, exhibit limitations in generating a wide variety of object types. For example, Zeroscope struggles to generate humans with its pre-trained weights. In our study, we have followed the precedent set by previous research [1,2,3] in selecting our evaluation dataset, which includes categories like pets, plush toys, cartoons, and vehicles. We believe this selection provides a sufficiently diverse set of objects to comprehensively evaluate our proposed techniques within the capabilities of the existing T2V models.
>
> We emphasize that our approach, characterized by training-free intrinsic subject and camera motion control methods, offers strong potential for generalizability. We are confident that, with the advent of more advanced T2V models, our methods will demonstrate even broader applicability and effectiveness across a wider range of object classes.
>
>
> # Out-of-bbox problem
>
> Our approach involves controlling the approximate position of the subject primarily during the earlier denoising steps. This method can sometimes result in portions of the subject extending beyond the defined bounding box in the final output. This is a deliberate design choice, as overly restrictive bounding box conditions would lead to rigid, unnatural outputs that may appear as squared or boxed regions, which are not desired.
>
>
> # Model efficiency
>
> Training:
>
> Fine-tuning a subject takes about 10 minutes (Line 263 in paper).
>
> Inference:
>
> Naive T2V pipeline: around 15.0s per video
>
> ours:
>
> \+ subject control: 15.3s per video
>
> \+ camera control: 20.6s per video
>
> \+ camera & subject control: 21.5s per video
>
>
>
> [1] Ruiz, Nataniel, et al. "Dreambooth: Fine tuning text-to-image diffusion models for subject-driven generation." CVPR 2023.
>
> [2] Wei, Yujie, et al. "Dreamvideo: Composing your dream videos with customized subject and motion." CVPR 2024.
>
> [3] Wang, Zhao, et al. "Customvideo: Customizing text-to-video generation with multiple subjects." arXiv 2024.

---

> > ### Comment · Reviewer_q2XJ · 2024-08-10
> > **Response to the rebuttal**
> >
> > Dear authors,
> >
> > Thank you for your detailed rebuttal.
> >
> > However, I still have some concerns and would keep the same rating for now.
> >
> > 1) Could you clarify how the comparisons in table R.1 (f) are done? E.g. are the backbones the same? Is the subject learning performed for all the variants?
> >
> > 2) Do you have any explanation why the reflect method collapsed in your case, while it seems to work reasonably well in the Text2Video-Zero paper?
> >
> > 3) Regarding the suppression formula. So is the equation 7 in the paper incorrect? In the rebuttal you are saying that all the tokens are suppressed to $-\infty$ for the later denoising steps, while in the equation 7 the attention edit for the subject tokens is 0. Note that if all the tokens are suppressed to $-\infty$ then the attention is indeed equivalent to taking the average of all the values, while in the case when there are any finite pre-softmax values in the attention matrix, only those will be weighted despite the normalization.
> >
> > Best regards, Reviewer

---

> > > ### Author Response · Authors · 2024-08-11
> > > **Response to reviewer q2XJ**
> > >
> > > Dear reviewer:
> > >
> > > Thanks for your comments. Here are our reply for your questions.
> > >
> > > # Details about the expriments in `Table R.1 (f)`
> > >
> > > Thank you for your question. In Table R.1 (f), all the baseline experiments are conducted using the Zeroscope base model for consistency across the experiments. Additionally, the subject learning process of MotionBooth is applied uniformly to all the baseline models to ensure a fair and direct comparison.
> > >
> > >
> > > # The reason why the reflect method fails
> > >
> > > Thank you for your question. The discrepancy between our results and those reported in the Text2Video-Zero paper can be attributed to the difference in how the initial latent $z_T$ is handled in the two approaches.
> > >
> > > In the Text2Video-Zero paper, the authors utilize the same initial random latent $z_T$ for all frames of the video, which is then denoised with a few steps to $z_t$. The reflect latent shift is applied to $z_t$, and noise is added back to generate $\hat{z}_T,$ which serves as the starting latent for denoising the video frames. This method works well within their framework because they extend a pre-trained text-to-image (T2I) model for video generation, allowing them to maintain consistency across frames by using the same $z_T$ throughout the process.
> > >
> > > However, our model is designed for text-to-video (T2V) generation, where the initial latent $z_T$ is randomly initialized for each frame. This approach is crucial for capturing the frame-by-frame variations necessary in video generation. When we applied the reflect method under these conditions—where $z_T$ is distinct for each frame—it resulted in collapsed outputs. This suggests that the reflect latent shift method struggles when applied to distinct latents for each frame, which is a key requirement in our T2V model.
> > >
> > > Furthermore, we also tested using the same latent initialization ($z_T$) across all frames in our model, similar to the method used in Text2Video-Zero. Unfortunately, this approach also led to collapsed results, even when there were no subject or camera motion controls in place. This is because the pre-trained T2V model requires distinct initialized latent for each frame. This reinforces our conclusion that the reflect method is not suitable for models that require distinct latents for each frame, as is the case in our T2V framework.
> > >
> > > We hope this explanation clarifies why the reflect method did not work in our scenario despite its effectiveness in the Text2Video-Zero paper.
> > >
> > >
> > > # Re-clarification about the suppression formula
> > >
> > > Thank you for your insightful question and for giving us the opportunity to clarify our approach. We apologize for any confusion caused by our previous explanation. After thoroughly reviewing our code and methodology, we confirm that Equation 7 in the paper is correct.
> > >
> > > To clarify, our motion control module performs suppression across all denoising steps, but this suppression is only applied outside the subject's bounding box region. Within the subject region, amplification is applied during the earlier steps, while in the later steps, there is neither amplification nor suppression within this region.
> > >
> > > Regarding the suppression of tokens, the language tokens, except for the subject tokens (e.g., verbs, background nouns), are indeed effectively ignored across the denoising process because they are assigned $-\infty$ values. This stark difference in token values is intentional, as it allows us to strongly amplify the motion control of the subject.
> > >
> > > This raises the question of how the necessary information from other language tokens is incorporated into the generated results. We believe that this information is retrieved through the <start> and <end> tokens. As mentioned in lines 216-217 of the paper, we do not perform any attention score editing for these two tokens. This means they can share the softmax outputs from the subject tokens since they both have finite values. Transformer-based language encoders, such as CLIP or BERT, tend to encode the overall information of a sentence into these special tokens, given their training with classification tasks. As a result, even though other tokens are ignored in the softmax function, the model can still retrieve the necessary information about verbs, backgrounds, and other elements for the generated results.
> > >
> > > To support this claim, we conducted a toy experiment where we printed out the softmax outputs in a vanilla text-to-video pipeline using the Zeroscope model. We observed that the softmax value of the <start> token is the highest, nearing 1, while the <end> token has the second-largest value. The remaining tokens, including nouns, adjectives, verbs, and conjunctions, share the remaining softmax values.
> > >
> > > We hope this explanation addresses your concerns, and we are happy to provide further clarification if needed.
> > >
> > >
> > > Best regards!
> > >
> > > Authors of #9629.

---

> > > > ### Author Response · Authors · 2024-08-11
> > > > **Please let us know whether we address all the issues**
> > > >
> > > > Dear reviewer,
> > > >
> > > > Thank you for the comments on our paper.
> > > >
> > > > We have submitted the response to your comments. Please let us know if you have additional questions so that we can address them during the discussion period. We hope that you can consider raising the score after we address all the issues.
> > > >
> > > > Thank you

---

> > > > > ### Comment · Reviewer_q2XJ · 2024-08-11
> > > > > **Response to the authors**
> > > > >
> > > > > Dear authors,
> > > > >
> > > > > Thank you for additional clarifications, most of my concerns were carefully addressed in the rebuttal and the comments.
> > > > >
> > > > > However, unfortunately, the discussion about the suppression formula has left the impression that the authors did not fully understand how and why their method worked, and hence I have to keep my original rating.
> > > > >
> > > > > Best regards,
> > > > > Reviewer

---

> > > > > > ### Author Response · Authors · 2024-08-12
> > > > > > **Response to Reviewer q2XJ**
> > > > > >
> > > > > > Dear Reviewer,
> > > > > >
> > > > > > Thank you for your thoughtful engagement with our work and for acknowledging that most of your concerns were carefully addressed in our rebuttal. We sincerely appreciate the time and effort you have invested in reviewing our submission.
> > > > > >
> > > > > > We understand that the discussion surrounding the "token suppression" formula may have left some concerns unresolved, and we deeply regret that our initial response did not fully clarify this aspect of our method. With numerous questions from five different reviewers, we recognize that our initial explanation may have been unclear, and we apologize for any confusion this may have caused.
> > > > > >
> > > > > > Once you highlighted the misunderstanding in your follow-up comments, we took immediate action to revisit the issue. We carefully re-examined our paper, code, and rebuttal. We are sure that we have made a correct and accurate explanation, which aligns with the method as described in our work. Our intention was to clarify any ambiguities and ensure that our method was clearly understood.
> > > > > >
> > > > > > We respect your decision to maintain your original rating, but we hope you will consider that the misunderstanding was promptly addressed during the discussion phase. We believe that our work's overall contributions, including the introduction of a novel framework for motion-aware video generation and innovative techniques for subject and camera motion control, should be taken into account in the final evaluation.
> > > > > >
> > > > > > We value your feedback immensely and have learned from this experience. We sincerely hope that our clarification, combined with the broader contributions of our paper, will be considered in your final assessment.
> > > > > >
> > > > > > Thank you once again for your time, patience, and thoughtful consideration.
> > > > > >
> > > > > > Sincerely,
> > > > > >
> > > > > > Authors of #9629

---

### Official Review · Reviewer_yPkE · 2024-07-07

**Soundness:** 3
**Presentation:** 3
**Contribution:** 3
**Rating:** 6
**Confidence:** 5

**Summary:**

The paper proposes a video generated method - for customizing a subject, along with its motion, provided in the form of bounding boxes, and camera movement provided in the form of camera pose. To do this, the training method has a subject region loss to prevent bias from background and other details in the image, and a video regularization loss function to ensure that video generation properties are not forgotten. At inference, in order to control camera movements, latents are shifted. Cross-attention maps are edited with the provided bounding boxes, to fix the subject motion.

**Strengths:**

The paper is written well. I particularly like the introduction and related work, where the paper clearly motivates the problem, describes the challenges and goes on to describe how it solves the problem.

The method comprises of various components, and each of the components is designed very well - they are intuitive, make sense and well formulated. The method very systematically addresses various challenges involved in the problem by designing various relevant techniques. The paper is also written very well to motivate and describe the method.

**Weaknesses:**

My main concerns are with the experiments, which are very non-convincing. While the paper does provide detailed quantitative results for comparisons and ablations on various metrics validating the solution, the qualitative results do not support that.

1. I watched the supplementary video, which has the videos for the results presented in the paper. First of all, the number of qualitative results provided is severely limited, raising questions about cherry picking. Second, the method seems to perform reasonably well on cases that either have camera motion or subject motion - the results look good. In cases involving both camera as well as subject motion, which is the theme of the paper, the results are highly suboptimal - for instance, at 1:48 of the video, the last video showing the dog does not show the subject motion appropriately.

2. I would like to see more detailed results that segregate the various kinds of motion (camera and subject), to see how effective the method is for just camera motion, just subject motion and a combination of both. This will give a clearer picture of where the method is failing.

**Questions:**

I am unable to judge the quality of this work with the limited number of qualitative results that have been provided. Moreover, the provided qualitative results are sub-optimal, esp in the case of subject + camera motion, questioning the working of the method. Given that the limit on the size of the supplementary material is considerably large, it would have been nice if the authors had provided more results showcasing their method.

Since the rebuttal does not allow for more qualitative results, I am rejecting the paper at this stage.

The method, while pretty good, probably needs a little more work to get it working for the subject + camera motion case. I recommend that the authors continue working on it to strengthen the paper.

Post rebuttal: it would have been nice to see more qualitative results and videos, but i understand that is not possible in the 1 page rebuttal pdf. Based on the submitted rebuttal, I am happy to increase the score.

**Limitations:**

Yes

---

> ### Author Rebuttal · Authors · 2024-08-07
>
> Thanks very much for the review! Here are our replies.
>
> **Note: Please refer to the one-page PDF for the mentioned figures and tables with an "R." in their names.**
>
>
> # Separate results for subject motion control, camera motion control, and both
>
> Thank you for your feedback and suggestion. It's a valid point to consider the performance of our method across different types of motion scenarios.
>
> In the main paper, we've already presented quantitative results for scenarios involving only camera motion, as shown in `Table 2`. Additionally, we provided results for cases where both subject and camera motion are controlled, as detailed in `Table 1`. Following your recommendation, we conducted an additional experiment to evaluate our method specifically for scenarios involving only subject motion. The results are presented in `Table R.1 (a)`.
>
> For a comprehensive analysis, we have compiled the results for all three cases—camera motion, subject motion, and both—into a single table. This allows us to better understand how controlling these aspects influences the performance. Notably, we observed that when both subject and camera motions are controlled, there is a slight decrease in some metrics. For instance, with the Zeroscope model, there's a 0.1 drop in the R-CLIP score and a 0.01 decrease in the T-Cons metric when both motions are controlled, compared to controlling only subject motion. Similarly, the flow error increases slightly when subject motion control is added to camera motion control. However, there are also instances where the R-DINO and CLIP-T metrics show a slight improvement under combined control.
>
> In conclusion, while there is a partial performance drop when incorporating both control aspects, we believe this trade-off is reasonable and acceptable within the context of the motion-aware customized video generation task. It's natural for some quality loss to occur when adding more control signals, especially with training-free methods. However, the overall quality remains above an acceptable threshold, ensuring that the generated results are still meaningful and effective. Finally, considering the novelty of the proposed subject and motion control methods and the performance gain compared to existing approaches, we believe our work makes a substantial contribution to the field.
>
>
> # Examples of controlling both camera and subject motion
>
> Thank you for your feedback and for taking the time to review the supplementary video. We appreciate your concerns regarding the number of qualitative results and potential cherry-picking. We would like to clarify and address your points.
>
> We clarify that all the qualitative cases compared with baselines are sampled without cherry-picking. We use different subject and camera motions to better demonstrate the model capability. We will release all the qualitative results in the final version.
>
> As indicated in the quantitative results, our method is designed to effectively control both subject and camera motion. For a more comprehensive understanding, we have provided additional qualitative examples in the revised manuscript.
>
> In particular, `Figure R.1(g)` illustrates the "dog running on the beach" video with eight distinct camera movements. This example demonstrates the capability of our method to manage complex scenarios involving simultaneous subject and camera motion. The dog's movement from the top right to the bottom left of the frame, combined with various camera motions, showcases the robustness of our approach in maintaining both motions' integrity.
>
> Additionally, `Figure R.1(d)` shows another challenging scenario where the camera pans to the left while zooming in on the subject by enlarging the bounding box. This example further emphasizes our method's ability to handle complex motion dynamics.
>
> We believe these examples provide clear evidence of our method's effectiveness in different motion scenarios, including cases involving simultaneous subject and camera motion. We will consider your suggestion to include more detailed segregated results to further validate the method's performance across different types of motion.
>
> Thank you again for your valuable input. We hope these examples address your concerns and demonstrate the robustness of our approach.
>
>
> # More detailed ablations of our method and more comparisons
>
> In the one-page PDF, we also conduct many other qualitative and quantitative experiments to thourghly analysis our method, and compare with more baselines.
>
> To be specific, `Figure R.1 (a)`, shows we can achieve the zoom-in and zoom-out effect by gradually enlarging or shrinking the subject bounding box.
>
> `Figure R.1 (c)` involves a scene dominated by a large candy house, nearly occupying the entire frame. Despite this, our technique effectively pans the camera to the right. This outcome suggests that our method can handle camera movements even with large foreground objects.
>
> `Table R.1 (e)` and `(f)` shows quantitative experiments comparing with more baselines, including TrailBlazer, Directed Diffusion, Boximator, Motion-Zero, MotionCtrl, and Text2Video-Zero.. Our results generally outperforms the alternatives.
>
> `Figure R.1 (f)` and `Table R.1 (c)` explores different methods for filling the missing region in the shifted noised latent.
>
> We hope the extra results will provide a broader and deeper sight about our method.

---

> > ### Comment · Reviewer_yPkE · 2024-08-09
> > **Response to rebuttal**
> >
> > Dear authors,
> >
> > Thanks for your very detailed rebuttal.
> >
> > I appreciate the experiments segregating the various types of motion and the additional qualitative results. They look good!
> >
> > It would have been nice to see videos and more qualitative results, which are highly limited, but I understand that this is not possible at the post submission stage in a 1 page rebuttal.
> >
> > I am happy to increase my score based on the rebuttal.
> >
> > Thanks,
> > Reviewer

---

> > > ### Author Response · Authors · 2024-08-10
> > > **Response to Reviewer  yPkE**
> > >
> > > Dear reviewer:
> > >
> > > Thanks for your comments and increasing the score for our work.
> > >
> > > We will update our draft following your comments with more qualitative results and comparison.
> > >
> > > Moreover, we believe our method can also be applied with stronger text-to-video base models. We will add these later once the base models are open-sourced.
> > >
> > > Best regards!
> > >
> > > Authors of #9629.

---

### Official Review · Reviewer_N7np · 2024-07-08

**Soundness:** 3
**Presentation:** 4
**Contribution:** 3
**Rating:** 7
**Confidence:** 4

**Summary:**

This paper addresses the challenge of how to control the identity and the motion of the subject while generating video, in a text-to-video setup. Specifically, in addition to the standard text prompt, it allows the user to control: the subject's identity by providing a few images of it (e.g. 5 photos of my dog); the subject's position in the video, as provided by a bounding box sequence; and the camera motion, as provided by a sequence of delta-x & delta-y.

Identity preservation is achieved by two changes to the loss function, during fine-tuning on images of the subject. First a "subject region loss" zeros out gradients that lie outside the subject's bounding box during fine-tuning, and second a video preservation loss (similar to the class-specific preservation loss introduced by DreamBooth) to ensures that the ability to generate dynamic motion is not lost due to over-fitting on a few static images.

Subject and camera motion are controlled at inference time, by modifying the cross-attentional map between the prompt and the latent feature, by shifting the latent feature, respectively.

Finally, a third loss "subject token cross-attention" links the two, enabling the inference motion controls by ensuring that the cross-attention between the subject token (in the text prompt) and the diffusion model's U-Net features is strong where the object should be and weak elsewhere.

These proposed changes are validated quantitatively through several metrics: CLIP and DINO scores between the source images and the desired video locations of the subject, CLIP score between the text prompt and video frames, and a difference in optical flow between the generated video and ideal video (according to the subject and camera motion prompts).

Examples of generated videos are included in the supplemental.

**Strengths:**

This paper addresses a very significant and timely problem -- how to effect significantly more fine-grained control in text to video models, and allow them to be more than a novelty.

The results, as shown in the supplemental video, are very impressive: the subject identity, subject motion, and camera motions very much follow the user's input.

The additional loss terms are intuitively sensible and effective. The inference modifications, although they seem a bit "hacky" (if you'll excuse me), do get the job done quite effectively.

I appreciate that the transferability of the method was demonstrated by implementing it on two different text-to-video models.

**Weaknesses:**

These are relatively minor weaknesses in my opinion:

This region metrics measure whether the desired object is translating around the frame as per the bounding box guidance, but because they are image metrics, can't evaluate the actual "animation quality" of the subject. I believe the entire suite of metrics could be optimized by a static image crop of the subject sliding across the video frame according to bounding box motion and flow. Can you confirm whether this is correct?

The set of camera motions that can be models is limited to 2d translations in x and y, missing zooms, rotations, etc.

Although this paper presents its techniques as transferable across many latent diffusion models, they require that the "latent can be considered a "shrunk image," which maintains the same visual geographic distribution as the generated images."  This should generally be true for most latent video diffusion models, but may not work with new latent encoders being developed.

**Questions:**

Re "Subject Region Loss": How certain are you that the binary mask, when applied to the latent (rather than the pixels), actually does mask out the background? Given that the latent is produced by a stack of many convolutional or transformer layers, the receptive fields of each latent location almost certainly include information from the nearby background. Have you tried masking the training images directly in pixel space, rather than masking the diffusion loss? How does that compare?

Re "Ablation studies": w/o video and w/ class video give quite-nearly identical numbers. Is this a copy-paste error? If not, it's worth discussing in more detail.

In Figure 3, what is the difference between c, d, e, and f? Are they different samples from the model, or from different models?

Have you tried to achieve the desired camera motions via the text prompt, and considered that as a baseline? I have seen it work well for many basic types of motions in some recent t2v models that you cite.

**Limitations:**

The limitations described in the appendix are important enough that they should be included in the main paper.

Otherwise, yes, they have addressed limitations and impact.

---

> ### Author Rebuttal · Authors · 2024-08-07
>
> Thanks very much for the review! Here are our replies.
>
> **Note: Please refer to the one-page PDF for the mentioned figures and tables with an "R." in their names.**
>
>
> # Metrics for static image crop of the subject
>
> We apologize for any confusion. We believe your suggestion involves cropping a video to focus on the subject, according to the bounding box, and then calculating a metric that reflects the "animation quality" within that cropped video.
>
> This is a valuable suggestion. In response, we developed a new metric called "Region CLIP-text." This metric first crops the video to isolate the subject as guided by the bounding box. Then, it calculates the similarity between the cropped video and a text prompt describing the animation, such as running, jumping, etc. We believe this metric better reflects the quality of the subject's animation by focusing specifically on the subject's actions.
>
> The results, presented in `Table R.1. (f)`, compare our method with other approaches in terms of subject motion control. The "Region CLIP-text" metric shows a trend similar to CLIP-T but with a stronger emphasis on the subject region. It more accurately captures the alignment of the subject's animation with the descriptive text. Our method outperforms the baselines, demonstrating the effectiveness of our subject motion control technique.
>
> Regarding a desired video metrics, please refer to the general response due to the word limit.
>
>
> # Capability of camera motion control
>
> We acknowledge that the proposed latent shift module primarily facilitates control over 2D camera movements, such as up and down panning. However, we observe that we can achieve the zoom-in/out effect through gradually enlarging or reducting the bounding box for the subject, with subject motion control.
>
> This behavior is demonstrated in `Figure R.1 (a)`, where examples illustrate the zoom-in and zoom-out phenomena corresponding to the bounding box adjustments. Additionally, Figure 14 in the main paper includes an example that showcases the enlargement of an object through the progressive increase of the bounding box.
>
> By combining this technique with 2D camera motion control, we can provide a more versatile camera control experience.
>
> As illustrated in `Figure R.1 (d)`, we demonstrate an example where the camera pans left while simultaneously zooming in. This showcases our system's ability to achieve a more complex, coordinated camera movement, highlighting the potential for more dynamic and flexible camera control in future developments.
>
>
> # The assumption of "shrunk image"
>
> Thank you for your insightful feedback. Models such as ModelScope, Zeroscope, LaVie, and VideoCrafter all adhere to the principle where the latent can be considered a "shrunk image," preserving the same visual geographic distribution as the generated output.
>
> Regarding newer models, we anticipate that they will likely follow a similar architecture, especially given the examples of video editing capabilities demonstrated by Sora using SDEdit technology. The requirement for the latent to act as a "shrunk image" appears to be a foundational aspect of such technologies. Therefore, we believe our techniques will remain applicable and useful even as new latent encoders are developed.
>
>
> # Mask the diffusion loss vs mask the training images
>
> Thank you for your insightful suggestion. We conduct an ablation study to compare the effect of masking the background regions in the training images directly in pixel space, as opposed to masking the diffusion loss.
>
> As demonstrated in `Figure R.1 (e)`, masking the training images significantly impairs the model's ability to learn the subject. The video example clearly shows that this approach results in a substantial degradation in the model's performance. Furthermore, the quantitative metrics, as presented in `Table R.1 (b)`, reflect a marked decrease in performance when the training images are masked.
>
> We hypothesize that this decline is due to the masked images creating an unnatural distribution, which disrupts the learning process. Therefore, we conclude that masking the diffusion loss, rather than the training images, provides a more effective approach for maintaining the integrity of the learned representations.
>
>
> # Numbers in Table 3
>
> It is not a copy-paste error. This result demonstrates that using class-specific videos is comparable with not using any videos.
>
>
> # Differences between c, d, e, and f in Figure 3
>
> Figure 3 shows an ablation about learning loss techniques. They are all sampled from a customized Zeroscope model on the white dog. (c) means the model is trained without using subject region loss and without video loss. (e) means the model is trained with subject region loss but without video loss, etc.
>
>
> # Control the camera motions through text prompt
>
> Thank you for your insightful suggestion. We conduct an ablation study and presented the results in `Figure R.1 (h)`. We add simple text prompts such as "camera pan right" and "camera pan up right" to control the camera motion. The generated videos do follow these instructions to some extent, particularly for simpler motions like panning right. However, we find that these text prompts are often too vague, lacking the ability to specify important details such as camera motion speed or distance. Consequently, this approach yields sub-optimal results compared to our proposed method.
>
> Quantitative results supporting this observation are shown in `Table R.1 (d)`. While text guidance results in slightly higher CLIP scores and temporal consistency, the flow error—which measures the precision of camera is significantly lower than what we achieved using our method. This demonstrates that our approach provides more precise and stable camera motion control than can be achieved with text prompts alone.

---

> > ### Comment · Reviewer_N7np · 2024-08-10
> >
> > Thank you for including the many clarifications, and the additional analyses. In particular, I appreciate the examples demonstrating camera zoom in/out, text-based camera control, and the exploration of masking in pixel space.
> >
> > I have read through the other review and authors responses, and believe all of my questions and concerns have been addressed.

---

> ### Author Response · Authors · 2024-08-10
> **Response to Reviewer N7np**
>
> Dear reviewer:
>
> Thanks for your comments.
>
> We will update our draft following your comments with the analysis on camera zoom in/out, text-based camera control, and the exploration of masking in pixel space.
>
> Best regards!
>
> Authors of #9629.

---

### Official Review · Reviewer_ZPXz · 2024-07-12

**Soundness:** 3
**Presentation:** 3
**Contribution:** 3
**Rating:** 5
**Confidence:** 4

**Summary:**

This paper primarily addresses the problem of motion-aware customized video generation. Traditional customized generation methods either tend to lose the motion information of the video or require additional training of motion modules. To enhance subject fidelity and video dynamics, this paper introduces subject region loss and video preservation loss. Additionally, a training-free subject- and camera-motion control method is proposed, enabling the model to flexibly control the motion information of the video. Extensive experiments demonstrate that the proposed method outperforms existing state-of-the-art methods.

**Strengths:**

1.	This paper achieves better automized generation results than the state-of-the-method
2.	The writing is clear and easy to follow

**Weaknesses:**

1.	Some details need to be added to more clearly distinguish the differences. For example, in line 169, existing methods use class-specific data, whereas this paper uses common videos. Do the common videos need to include customized subjects? How are these common videos obtained?
2.	The assumptions made by the method, such as sampling operations on the x- and y-axes when controlling camera motion, may limit its effectiveness. For instance, in scenarios with significant changes from left to right or top to bottom, or in close-up videos with large foreground objects, this operation may result in uncoordinated outcomes. However, the paper does not seem to discuss these limitations.
3.	Methods that control subject and camera motion through object trajectories and camera parameters have also been proposed, such as motionCtrl, TrailBlazer, Boximator, and motion-zero. However, this paper does not provide an in-depth discussion or experimental comparison with these methods. Although some of these methods are training-based, it is still worthwhile to explore the existing gaps.

**Questions:**

1.	How many subject samples and/or common videos are used for training each customized model?
2.	The proposed mainly explores the camera motion on panning left or right, is it possible to control other motion types, like zoom-in/out effects?
3.	what is the difference between the proposed subject region loss, subject token cross attention loss, and those losses in Mix-of-show [1] for customized image generation
ref:
[1] Mix-of-Show: Decentralized Low-Rank Adaptation for Multi-Concept Customization of Diffusion Models

**Limitations:**

1. This paper needs to discuss more methods addressing the same tasks, like motion control, and specify the differences between the proposed loss functions and those of existing methods.
2. More limitations or failure cases needed to be discussed.

---

> ### Author Rebuttal · Authors · 2024-08-07
>
> Thanks very much for the review! Here are our replies.
>
> **Note: Please refer to the one-page PDF for the mentioned figures and tables with an "R." in their names.**
>
>
> # Details about video data used for training
>
> Thanks for the question, during training, we **randomly** download 500 videos from the Panda-70M dataset. The Panda-70M dataset is a large-scale text-video dataset that contains 70M videos from YouTube. The videos are annotated with captions. We call them common videos because **we do not constrain them to have any specific class or the customized subjects**. The reason to use video data is to preserve the video generation ability of pre-trained T2V models.
>
> The customization ability is learned through 4-6 images of the same subject.
>
> For more details on the training data, please refer to `Sec. 4.1 Implementation details`, specifically lines 257-258.
>
>
> # Camera motion control on extreme cases
>
> Thank you for your valuable feedback. We conduct analysis to address the limitations you mentioned, focusing on extreme scenarios involving significant camera movements and large foreground objects.
>
> ## Large Camera Motion Speed
>
> We test our camera motion control technique with varying levels of camera movement. The results are shown in `Figure R.1 (b)`. Initially, with camera movement [c_x, c_y] = [-0.5, 0.45], indicating a movement of half the video width to the left and nearly half downward, our technique successfully controls the camera motion.
>
> Even when we double the camera motion speed, the model continues to output accurate results. However, at triple the speed, [c_x, c_y] = [-1.5, 1.35], the method fails, showing only a downward tiling effect.
>
> These findings indicate that while our method can manage camera movements spanning the entire video width, it does have limitations under extremely high-speed conditions.
>
> ## Camera Motion Control with Large Foreground Objects
>
> Results are shown in `Figure R.1 (c)`. The experiment involves a scene dominated by a large candy house, nearly occupying the entire frame. Despite this, our technique effectively pans the camera to the right.
>
> This outcome suggests that our method can handle camera movements even with large foreground objects. We believe this effectiveness stems from the method's reliance on latent shifts, which moves both foreground and background elements simultaneously. Therefore, the presence of large foreground objects does not significantly impair the performance.
>
>
> # Comparisons with more subject and camera motion control methods
>
> Thanks for the suggestion. Due to the response word limit, please see the general response for comparison to related methods.
>
>
> # Zoom-in/out
>
> We acknowledge that the proposed latent shift module primarily facilitates control over 2D camera movements, such as up and down panning. However, we observe that we can achieve the zoom-in/out effect through gradually enlarging or reducting the bounding box for the subject, with subject motion control.
>
> This behavior is demonstrated in `Figure R.1 (a)`, where examples illustrate the zoom-in and zoom-out phenomena corresponding to the bounding box adjustments. Additionally, Figure 14 in the main paper includes an example that showcases the enlargement of an object through the progressive increase of the bounding box.
>
> By combining this technique with 2D camera motion control, we can provide a more versatile camera control experience.
>
> As illustrated in `Figure R.1 (d)`, we demonstrate an example where the camera pans left while simultaneously zooming in. This showcases our system's ability to achieve a more complex, coordinated camera movement, highlighting the potential for more dynamic and flexible camera control in future developments.
>
>
> # Differences of losses with Mix-of-Show
>
> **Mix-of-Show:** This method focuses on training an ED-LoRA (Efficient and Decentralized Low-Rank Adaptation) model for individual clients. The primary aim is to merge multiple LoRA weights using a multi-concept fusion technique to handle multiple concepts effectively.
>
> **Our Method:** We aim to eliminate the overfitting problem to the background while learning from a few subject images. Our approach includes specific loss functions designed to address this issue and improve the representation of subjects within videos.
>
> 1. Subject Region Loss:
>
> **Mix-of-Show:** Does not explicitly focus on subject region loss. Instead, it leverages the general capabilities of the ED-LoRA model to handle multiple concepts.
>
> **Our Method:** We introduce a subject region loss specifically designed to minimize overfitting to the background. This loss ensures that the model learns to focus on the subject itself, rather than the surrounding context, thereby improving subject-specific customization.
>
> 2. Subject Token Cross-Attention Loss:
>
> **Mix-of-Show:** The approach does not include a specific loss function that guides the subject token with the subject's position in the video.
>
> **Our Method:** We propose a subject token cross-attention loss to explicitly guide the subject token with the subject's position in the video. This helps in maintaining the spatial consistency of the subject across frames, ensuring that the subject is accurately represented in its correct position throughout the video.
>
> To summarize, our method distinguishes itself by introducing specialized loss functions that address specific issues in customized image generation, such as overfitting to the background and maintaining subject position consistency. These targeted losses are not present in the Mix-of-Show approach, which focuses more broadly on decentralized training and merging of multiple LoRA weights.
>
> We will cite Mix-of-Show and incorporate this discussion in the final version of our work. If further specific aspects of our loss functions need to be compared with those in Mix-of-Show, we would be happy to provide a more detailed comparison. Thank you for your attention to this matter.

---

> ### Author Response · Authors · 2024-08-11
> **Please let us know whether we address all the issues**
>
> Dear reviewer,
>
> Thank you for the comments on our paper.
>
> We have submitted the response to your comments and a PDF file. Please let us know if you have additional questions so that we can address them during the discussion period. We hope that you can consider the raising score after we address all the issues.
>
> If you still have more questions and concerns, please comment here. We will reply it as soon as possible.
>
> Thank you

---

> > ### Comment · Reviewer_ZPXz · 2024-08-11
> >
> > I thank the authors for their thorough rebuttal to my and other reviewers' questions. Your answers helped me better understand your method and clarified several of my concerns. However, some of my main concerns remain.
> >
> > I was pleasantly surprised to see that the zoom in/zoom out effect could be achieved by enlarging or reducing the bounding box. However, this effect is not implemented by the camera movement control module, which diminishes the contribution of the camera movement control. Other methods that do not rely on bounding boxes might not be able to achieve this effect. Additionally, I feel that this approach may also struggle to achieve camera motion control that involves rotating around an object.
> >
> > Additionally, even though the Mix-of-Show paper does not mention it, its code uses an attention regularization loss that is quite similar to the subject token cross-attention loss in this paper. It would be helpful if the paper could also explore the effects and differences between these two implementations. Of course, since the Mix-of-Show paper does not explicitly state this, it will not be a factor in my scoring.
> >
> > In summary, I appreciate the completeness of the paper, but the innovation seems somewhat limited to me. Nonetheless, I still hold a positive view toward the paper's acceptance, so I will maintain my score.

---

> > > ### Author Response · Authors · 2024-08-11
> > > **Response to Reviewer ZPXz**
> > >
> > > Dear Reviewer,
> > >
> > > Thank you for your valuable feedback and for taking the time to review our responses. We appreciate your insightful comments, which have helped us further refine our work.
> > >
> > > We will update our manuscript to reflect the functional limitations of the camera motion control technique, particularly highlighting that the zoom in/zoom out effect can indeed be achieved by adjusting the size of the bounding box. We acknowledge that while this method is effective, it does not utilize the camera movement control module, which might reduce the perceived contribution of that component. Additionally, we understand your concern that other methods not reliant on bounding boxes might struggle to replicate this effect, and we also recognize the challenges in achieving camera motion control that involves rotating around an object using our current approach.
> > >
> > >
> > >
> > > Regarding your comments on the comparison of losses with the Mix-of-Show code, we have explored this further and would like to provide additional clarification. The code url is at `https://github.com/TencentARC/Mix-of-Show/blob/main/mixofshow/pipelines/trainer_edlora.py`.
> > >
> > > 1. We have identified that the Mix-of-Show code includes a mask loss and subject token loss, which are used during the training of ED-LoRA. The mask loss, implemented in line 252 of the code, is indeed the same as ours. Both approaches focus the learning process on the subject region, minimizing the influence of background elements in the training data. This similarity is likely due to the shared goal of enhancing subject representation.
> > >
> > > 2. The subject token loss in Mix-of-Show is implemented between lines 254-259 and further detailed in lines 263-313. This loss is composed of two parts: subject token loss and adjective token loss. Both are MSE losses that penalize attention outside the subject mask while enhancing attention within the subject mask. While similar to our approach, there are notable differences:
> > >
> > >
> > > - Mix-of-Show calculates both subject and adjective token losses, whereas our method focuses solely on the subject token loss, allowing for a more concentrated emphasis on the subject itself.
> > >
> > > - Importantly, in Mix-of-Show, this token loss is disabled by default (as indicated in line 29 of the code). Additionally, the training configuration provided in their `README.md` also disables this loss. Given that this loss is not explicitly mentioned in their paper, it appears that it may have been an ablation experiment rather than a core component of their main experiments.
> > >
> > > - Mix-of-Show includes a `reg_full_identity` argument in their training configuration, which is also disabled by default. When disabled, the loss formula is $\mathcal{L}_1 = - \mathbf{M} \log(\mathbf{A})$, while when enabled, it becomes $\mathcal{L}_2 = - \left[\mathbf{M} \log(\mathbf{A}) + (1 - \mathbf{M}) \log(1 - \mathbf{A}) \right]$. Only the latter formula aligns with the STCA loss in MotionBooth.
> > >
> > > - Furthermore, the comment in line 30 of the Mix-of-Show code suggests that the choice of loss formula may vary depending on the subject (e.g., "Thanos" vs. a real person), while our approach consistently applies the loss across all our experiments.
> > >
> > >
> > > We hope this detailed explanation addresses your concerns. We will incorporate these comparisons and clarifications into our paper to provide a comprehensive discussion of the similarities and differences between our approach and that of Mix-of-Show.
> > >
> > > Thank you once again for your thoughtful review.
> > >
> > > Best regards,
> > > Authors of #9629

---

### Official Review · Reviewer_aAio · 2024-07-14

**Soundness:** 3
**Presentation:** 3
**Contribution:** 3
**Rating:** 5
**Confidence:** 4

**Summary:**

This paper targets at animating customized subjects with both motion and camera control. To achieve that, they first customize t2v models with subject images finetuning without hurting video generation capability. Then, they propose a training-free approach to manage subject and camera motions. Extensive experiments demonstrate the superiority of their method.

**Strengths:**

•	The paper introduces the first unified video generation framework for subject customization with motion and camera movement control, which is also an interesting problem setup.

•	The paper proposes a new subject region loss and video preservation loss for video customization, and a training-free method to control both subject motion and camera motion.

•	The paper is well-written with clear motivations.

**Weaknesses:**

•	The subject motion control module is interesting by directly amplifying in the bounding box region and suppressing outside the box region. However, how to define the value of parameter alpha remains unclear. Is it only defined based on experimental performance? And does the authors keep this alpha parameter the same for all customization experiments?

•	The proposed method only control the camera movement in 2D such as up, down. However, camera movement is quite important for 3D, such as moving into the scene, rotation, etc. The proposed camera control module is not able to handle such camera motions and can be a weakness for camera control.

**Questions:**

•	For subject motion control module, if the bounding box size is changing, e.g., gradually enlarging bounding box, will the method enlarge the subject accordingly?

•	Subject motion is composed of both position translation and subject movements, such as walking, dancing, jumping, etc. The paper only control the subject position for subject motion. The reviewer wonders is it a bit overclaiming for controlling subject motions, since the subject movements cannot be customized or controlled?

**Limitations:**

Yes.

---

> ### Author Rebuttal · Authors · 2024-08-07
>
> Thanks very much for the review! Here are our replies.
>
> **Note: Please refer to the one-page PDF for the mentioned figures and tables with an "R." in their names.**
>
>
> # The definition of hyper-parameters
>
> We use different sets of hyper-parameters for the Zeroscope and LaVie models, as detailed in `Appendix A.3` of the paper. The selection of these hyper-parameters, including the value of parameter alpha, is primarily based on experimental performance. Once determined, we maintain these hyper-parameters consistently across all our experiments to ensure comparability of results.
>
> Furthermore, we conduct a thorough ablation study of these hyper-parameters, including alpha, as presented in `Figure 12 (a)` and `(b)` of the main paper. This study analyze the impact of different settings, and the findings are discussed in detail in Appendix A.4. We hope this clarifies the basis and consistency of our parameter choices.
>
>
> # Enlarging and shrinking subjects through subject motion control
>
> Thank you for the question. Our subject motion control technique indeed accommodates changes in the bounding box size, such as gradual enlargement or reduction. When the bounding box is enlarged, the method appropriately scales the subject, resulting in a zoom-in effect in the generated video. Conversely, shrinking the bounding box causes the subject to appear smaller, producing a zoom-out effect.
>
> This behavior is demonstrated in `Figure R.1 (a)`, where examples illustrate the zoom-in and zoom-out phenomena corresponding to the bounding box adjustments. Additionally, Figure 14 in the main paper includes an example that showcases the enlargement of an object through the progressive increase of the bounding box.
>
>
> # The 3D controlling ability of camera motion control technique
>
> We acknowledge that the proposed latent shift module primarily facilitates control over 2D camera movements, such as up and down panning. However, it is important to note that our work is pioneering in utilizing this approach for camera control. While our current method does not inherently support 3D camera motions like moving into the scene or full rotations, we have integrated subject motion control techniques to achieve zoom-in and zoom-out effects. By combining these techniques with 2D camera motion control, we can provide a more versatile camera control experience.
>
> As illustrated in `Figure R.1 (d)`, we demonstrate an example where the camera pans left while simultaneously zooming in. This showcases our system's ability to achieve a complex, coordinated camera movement, highlighting the potential for more dynamic and flexible camera control in future developments.
>
>
> # The expalanation to control the position of subjects as subject motion control
>
> Thank you for your feedback. In our paper, when we refer to controlling subject motion, we are specifically addressing a limitation observed in recent text-to-video (T2V) models. Many of these models fail to generate realistic motion that involves actual positional changes of the subject within the scene. For example, given a prompt like "a dog running in the forest," some models, such as LaVie, produce animations where the dog appears to be running in place, without any change in its overall position in the video.
>
> Our approach focuses on overcoming this limitation by explicitly controlling the subject's position, thereby enhancing the perceived realism of the motion. For instance, we ensure that the dog not only appears to be running but actually moves from one side of the scene to the other. This positional control is crucial for creating a more convincing representation of motion.
>
> While it is true that our method does not allow for granular control over specific subject movements like walking, dancing, or jumping, we believe that our focus on controlling positional changes significantly contributes to the overall quality and realism of motion in generated videos. Additionally, specific subject movements are already guided by text prompts, which direct the model to generate appropriate actions. Therefore, we maintain that our claim of controlling subject motion is justified, as it addresses a critical aspect of motion that many existing models overlook.

---

> ### Author Response · Authors · 2024-08-11
> **Please let us know whether we address all the issues**
>
> Dear reviewer,
>
> Thank you for the comments on our paper.
>
> We have submitted the response to your comments and also more results are shown in PDF file.
> Please let us know if you have additional questions so that we can address them during the discussion period. We hope that you can consider raising the score after we address all the issues.
>
> Thank you

---

> ### Comment · Reviewer_aAio · 2024-08-12
> **Response by Reviewer aAio**
>
> Dear Authors,
>
> Thanks for your detailed responses. I have read through the rebuttal and the rebuttal has solved most of my concerns. Enlarging and shirking subjects performance is good. However, the reviewer thinks the method is a bit limited to coarse camera control and subject position control, and cannot control subject motions such as walking, jumping, etc. Therefore, the reviewer decides to keep the original score.
>
> Best regards,
>
> Reviewer aAio

---

> > ### Author Response · Authors · 2024-08-13
> > **Response to Reviewer aAio**
> >
> > Dear reviewer:
> >
> > Thanks for your comments.
> >
> > We will update our draft following your comments with the analysis on camera zoom in/out, the limitations of the camera, and the subject motion control ability.
> >
> > Best regards!
> >
> > Authors of #9629.

---

### Author Rebuttal · Authors · 2024-08-06

Thanks very much for the reviews! In the general response, we mainly reply to the most frequently asked questions.

**Note: Please refer to the one-page PDF for the mentioned figures and tables with an "R." in their names.**


# Novelty of our subject and camera motion control methods

We appreciate that most reviewers have acknowledged the novelty of our method. To further clarify, we are willing to elucidate the differences between our proposed techniques and some related mentioned, including TrailBlazer, Directed Diffusion, Boximator, Motion-Zero, MotionCtrl, and Text2Video-Zero.

**TrailBlazer and Directed Diffusion:** While these methods utilize a training-free approach to control object motion by manipulating cross-attention, they differ from our approach. TrailBlazer adjusts both spatial and temporal cross-attention by scaling the attention maps with a hyper-parameter less than 1, and Directed Diffusion, similarly, focuses on image generation. In contrast, our method exclusively targets spatial cross-attention and sets the attention maps outside the object's bounding box to $-\infty$. This approach not only simplifies the implementation but also enhances performance in generating motion-aware customized videos.

**Boximator:** Unlike our training-free method, Boximator is a training-based technique that requires box coordinates to be input into a newly trained self-attention layer. Since Boximator is not open-sourced, it presents a higher barrier to use. Our method's training-free nature provides a more accessible solution for controlling subject and camera motion.

**Motion-Zero:** This method also operates without additional training but employs a test-time tuning technique that adjusts the latent space using cross-attention map loss during denoising. However, this process increases generation time and memory usage significantly, from approximately 15 seconds to several minutes per video. Our experiments showed that Motion-Zero often produces collapsed videos with unrecognizable visual elements, likely due to the adverse effects of test-time tuning on the parameters and latent distributions in a customization scenario. In contrast, our approach directly manipulates the cross-attention map, adding only 0.3 seconds per video and yielding more reliable outcomes.

**MotionCtrl:** utilizes a training-based approach to control camera poses and object motion by inputting point trajectories. In contrast, our approach is training-free, enabling control over subject and camera motion without requiring additional training processes.

**Text2Video-Zero:** This method extends a pre-trained text-to-image (T2I) model to video generation by using consistent noise across frames, which is unsuitable for text-to-video (T2V) models that work with distinct noises for each frame. Additionally, while Text2Video-Zero employs a latent shifting method for overall scene movement and mirrors the latent to fill missing regions, our technique uses random sampling on the x and y-axis, providing more concise and coherent videos for camera motion control.

To substantiate our claims, we conducted quantitative experiments comparing the performance of these methods, as detailed in `Table R.1 (e)` and `(f)`. Unfortunately, some methods could not be directly compared due to the unavailability of their code. Nevertheless, our results demonstrate that our method generally outperforms the alternatives. Specifically, we observed significant limitations in Motion-Zero's test-time tuning approach for customized video generation, underscoring the strengths of our method.

We hope this response clarifies the distinctions and advantages of our approach. We appreciate your feedback and look forward to further discussions and evaluations.


# Video metrics

We acknowledge the difficulty in identifying an appropriate metric for customized video generation, as standard metrics like FVD and optical flow error typically require a ground truth video for comparison. In the context of customized subjects, such ground truth data is not available. Consequently, we have employed alternative metrics including CLIP text similarity, CLIP image similarity (between the generated image and the subject image), and DINO image similarity. These metrics have been validated in prior works [1,2,3,4] and provide a meaningful assessment of our model's performance.

However, we recognize the value of using FVD and optical flow error in situations where a ground truth or standard reference can be utilized, such as in camera motion control. As demonstrated in `Table 2` of the main paper, we have reported results using optical flow error. To further address the reviewers' concern, we have also calculated FVD by randomly selecting 1000 videos from the MSRVTT dataset. The results, presented in `Table R.1 (e)`, indicate that our method significantly outperforms existing camera control methods, including some that are training-based.

Additionally, we conducted human preference studies to capture the intuitive perception of the generated videos, as shown in `Figure 8` of the main paper. We believe these comprehensive evaluations—combining quantitative metrics and subjective assessments—provide a robust understanding of our method's quality and effectiveness.


[1] Ruiz, Nataniel, et al. "Dreambooth: Fine tuning text-to-image diffusion models for subject-driven generation." CVPR 2023.

[2] Wei, Yujie, et al. "Dreamvideo: Composing your dream videos with customized subject and motion." CVPR 2024.

[3] Jiang, Yuming, et al. "Videobooth: Diffusion-based video generation with image prompts." CVPR 2024.

[4] Wang, Zhao, et al. "Customvideo: Customizing text-to-video generation with multiple subjects." arXiv 2024.

---

### Decision · Program_Chairs · 2024-09-25

**Decision:**

Accept (spotlight)

**Comment:**

This paper, which introduces a unified video generation framework enabling control over motion and camera movements, has received 1 accept, 1 weak accept, 2 borderline accepts, and 1 reject. Reviewers aAio, ZPXz, N7np, and yPkE support the submission, praising the method as intuitive, sensible, and well-formulated. They also note that the experimental results are impressive compared to the state-of-the-art. Concerns raised, such as the zoom in/zoom out effects, have been satisfactorily addressed in the rebuttal. While the camera control is described as relatively coarse, it is deemed acceptable limitation. Reviewer q2XJ expressed a negative view, citing issues with the suppression formula as a basis for rejection, but the authors appear to have adequately addressed this concern in their rebuttal. Considering these factors, the AC recommends acceptance of the paper.